# Migration arrest and transendothelial trafficking of human pathogenic-like Th17 cells are mediated by differentially positioned chemokines

Farhat Parween[1], Satya P. Singh[1,7], Nausheen Kathuria[1,7], Hongwei H. Zhang[1,7], Shinji Ashida[2,7], Francisco A. Otaizo-Carrasquero [3], Amirhossein Shamsaddini[3], Paul J. Gardina [3], Sundar Ganesan[4], Juraj Kabat[4], Hernan A. Lorenzi [5], Deanna J. Riley[6], Timothy G. Myers [3], Stefania Pittaluga [6], Bibiana Bielekova[2] & Joshua M. Farber [1] ✉

Human Th17/type 17 cells express the chemokine receptor CCR6, but the functions of CCR6 and other chemokine receptors in human type 17 Th cell extravasation have not been fully delineated. Here we show that human peripheral blood CD4⁺CCR6⁺ T cells co-expressing CCR2 have a pathogenic Th17 signature, can produce inflammatory cytokines without T cell receptor activation, and show enhanced expression of pathogenicity-associated and activation-associated genes in the cerebrospinal fluid of patients with multiple sclerosis as compared to controls. In flow chambers with activated endothelial cell (EC) monolayers, CD4⁺CCR6⁺CCR2⁺ T cells are efficient at transendothelial migration (TEM). Ligands for CCR5, CCR6 and CXCR3 localize to EC surfaces and mediate only arrest, whereas CCR2 ligands fail to bind well to ECs and mediate only TEM. Conversely, expressing a chimeric CCR2 ligand engineered to bind glycosaminoglycans on ECs results in CCR2-mediated arrest but blocks TEM induction. Our results from human pathogenic-like type 17 cells thus suggest that T cell migration arrest requires chemokine bound to EC surfaces, whereas TEM requires a transendothelial chemokine gradient.

The multistep model of migration of leukocytes across vascular barriers classically involves three sequential steps: (1) selectin-mediated rolling of leukocytes across endothelium, (2) chemoattractant-induced integrin-dependent firm arrest of leukocytes on endothelium, and (3) chemoattractant-dependent migration of leukocytes between adjacent (or through) endothelial cells (ECs) into subjacent tissue[1,2]. How specific adhesins and chemoattractants cooperate within this framework for extravasation of specific leukocyte subsets across specific

[1]Inflammation Biology Section, Laboratory of Molecular Immunology, National Institute of Allergy and Infectious Diseases, National Institutes of Health, Bethesda, MD, USA. [2]Neuroimmunological Diseases Section, Laboratory of Clinical Immunology and Microbiology, National Institute of Allergy and Infectious Diseases, National Institutes of Health, Bethesda, MD, USA. [3]Genomic Technologies Section, Research Technologies Branch, National Institute of Allergy and Infectious Diseases, National Institutes of Health, Bethesda, MD, USA. [4]Biological Imaging Section, Research Technologies Branch, National Institute of Allergy and Infectious Diseases, National Institutes of Health, Bethesda, MD, USA. [5]Bioinformatics and Computational Biosciences Branch, National Institute of Allergy and Infectious Diseases, National Institutes of Health, Bethesda, MD, USA. [6]Laboratory of Pathology, Center for Cancer Research, National Cancer Institute, National Institutes of Health, Bethesda, MD, USA. [7]These authors contributed equally: Satya P. Singh, Nausheen Kathuria, Hongwei H. Zhang, Shinji Ashida. ✉e-mail: jfarber@niaid.nih.gov

blood vessels into tissue has not been fully delineated in any species and is particularly poorly understood for effector/memory T cells and in humans.

Effector/memory T cells provide protection against infection but may also mediate damage in peripheral tissue[3]. Memory T cells can be classified according to multiple overlapping features, such as pathways of migration or patterns of cytokine production[4], and they include in addition to tissue-resident cells[5], circulating memory T cells, which migrate from blood into peripheral tissue and rapidly respond to invading pathogens[6]. Th17 cells express the signature transcription factor retinoid-related orphan receptor gamma-t (RORγt) and the signature cytokine IL-17A. Their function is to protect mucosal surfaces from extracellular pathogens. However, they may also mediate tissue damage in the context of autoimmune disease[7].

An integral component of Th cell physiology and pathogenicity is the ability to traffic into and within tissue. One relevant chemokine receptor, CCR6, is expressed on all human T cells that can produce IL-17A/F and the other proteins associated with the Th17 family, such as RORγt, IL-23R, IL-22 and CCL20, and we have used CCR6 to identify this greater population that we include in the category of type 17 cells[8–10]. Type 17 cells can also make effector cytokines that are not unique to this subset, such as the canonical Th1 cytokine, IFN-γ, and some type 17 cells have been characterized as pathogenic in mouse models of autoimmune disease through their production of IFN-γ and/or GM-CSF[7].

Consistent with its pattern of expression and activity, CCR6 has been shown to be important in mouse models of autoimmune disease such as experimental autoimmune encephalomyelitis (EAE) and psoriasis-like inflammation of the skin[11,12]. Whereas most chemokine receptors bind multiple chemokines and vice versa, CCL20 is the only chemokine ligand for CCR6 and CCR6 is the only known G protein signaling receptor for CCL20[13,14], although CCL20 also binds to the atypical receptor, ACKR4[15]. Redundancy in the activity of trafficking receptors may be important for extravasation of CCR6-expressing Th cells, since cells within the CCR6+ population express multiple other chemokine receptors, including CCR2, CCR4, CCR5, CXCR3, and CXCR4 in various combinations ([16] and this manuscript). In our previous work with human CD8α+ MAIT cells, we found subspecialized roles for multiple chemokine receptors within the multistep model of extravasation, whereby CCR6 mediated firm arrest on activated ECs and CCR2, a receptor studied primarily for its role in monocyte trafficking[17], was responsible for the step of transendothelial migration (TEM)[18].

Here we focus on understanding how the chemokine system contributes to the effector capabilities of human pathogenic-like type 17 cells through the chemokine-mediated critical steps that enable the trafficking of cells into inflamed tissue. We find that expression of CCR2 characterizes CCR6+/type 17 cells with pathogenic features in the blood and the cerebrospinal fluid (CSF), and that in flow chamber assays using CCR6+CCR2+ cells chemokines that bind well to the surfaces of activated ECs mediate T cell migration arrest through the activities of CCR5, CCR6, and CXCR3, whereas only CCR2, whose secreted ligands bind poorly to the EC surfaces, mediates TEM. Our data suggest that while T cell arrest requires EC-bound chemokine, TEM depends on a chemokine transendothelial gradient, which can be formed by removal of non-bound, luminal chemokine by vascular flow. Understanding the redundant and non-redundant activities of specific chemokines and their receptors in the trafficking of highly inflammatory human T cells, and the mechanisms underlying these activities, should allow for more informed approaches to blocking or enhancing T cell trafficking for therapeutic benefit.

## Results

### CCR2 identifies human type 17 cells with a pathogenic signature

To understand the mechanisms used by bona fide human memory type 17 Th cells to traffic into inflamed tissue, based on insights from our studies of CD8α+ MAIT cells, where we identified an important role for CCR2[18], we focused in the current work on resting, non-regulatory CD4+ effector/memory Th cells that co-express CCR6 and CCR2. Because we found that CCR2+ cells within the CCR6+ population were generally CCR6high, for purposes of comparison we used flow cytometry to separate CD4+ cells from healthy donors into naïve and HLA-DR−CD25−/lowCD127+ memory-phenotype populations and further separated memory cells into those that were CCR6−CCR2−, CCR6lowCCR2−, CCR6+(high)CCR2− and CCR6+(high)CCR2+ (Supplementary Fig. 1a and Fig. 1a). Particularly for CCR2, in which positive and negative cells do not form discrete populations, cells labeled as negative may nonetheless express low numbers of receptors.

We first defined CCR6+(high)CCR2+ and CCR6−CCR2− cells at the molecular level by global gene expression using bulk RNA-seq, analyzing both resting cells and cells pharmacologically activated ex vivo with PMA and ionomycin to induce cytokine gene expression. Gene Ontology (GO) analysis showed enrichment in CCR6+(high)CCR2+ cells for transcripts associated with leukocyte migration, adhesion, and differentiation (Fig. 1b). Type 17 genes such as IL17A, IL17F, IL22, CCL20 and RORC were more highly expressed in the CCR6-positive cells, consistent with what we and others have reported previously[8,9], particularly in the CCR6+(high)CCR2− and CCR6+(high)CCR2+ subgroups (Fig. 1c and Supplementary Fig. 2a). We noted that as compared with the other CCR6-expressing subgroups, the CCR6+(high)CCR2+ cells showed higher expression of genes linked to mouse models of autoimmune disease or a pathogenic profile in human Th17 cells, including RORC, IL23R, TBX21, IFNG, BHLHE40, CSF2 (encoding GM-CSF), IL1R1 and ABCB1[19,20]. Further, gene set enrichment analysis (GSEA) of the RNA-seq data from CCR6+(high)CCR2+ vs. CCR6−CCR2− cells showed a significant enrichment score against a gene set based on mouse data identifying a signature of pathogenic vs. non-pathogenic Th17 cells[19] (Fig. 1d). Relative levels of gene expression among the T cell subgroups did not differ significantly between the samples from the two donors, nor between the activated and non-activated cells (Fig. 1c and Supplementary Fig. 2a).

We next inspected expression of pro-inflammatory proteins of interest among chemokine receptors and cytokines. As shown in Supplementary Fig. 2b (with gating strategy in Supplementary Fig. 1b), the CCR6+(high)CCR2+ cells showed, in addition to CCR6 and CCR2, increased expression of the inflammatory chemokine receptors CXCR6, CCR5 and CXCR3. Expression of mRNA for these receptors generally matched the pattern of surface staining (Supplementary Fig. 2c). Intracellular staining for cytokines (and CCL20) in cells activated with PMA and ionomycin ex vivo (Fig. 1e, with gating strategy in Supplementary Fig. 1c) also matched the RNA-seq data. The CCR6+(high)CCR2+ subgroup contained the highest percentages of cells staining for the pathogenic cytokines IFN-γ, GM-CSF, and TNF-α, but not for IL-17A. We also considered, given our earlier data suggesting a first-responder phenotype for CD4+CCR2+ cells[16] whether the CCR6+(high)CCR2+ cells might produce effector cytokines after exposure to a pro-inflammatory cytokine stimulus in the absence of cognate antigen. As shown in Fig. 1f, among the subgroups of CD4+ T cells, the CCR6+(high)CCR2+ cells secreted the most IFN-γ and GM-CSF after ex vivo culture with combinations of inflammatory cytokines, similar to the results following pharmacologic activation. Together, the data suggest that within the CCR6+, type 17 memory population, CCR2 marks cells with increased pathogenic potential.

### Single-cell analysis of CCR6+(high)CCR2+ Th cells reveals heterogeneity in expression patterns for pathogenicity-associated genes

Heterogeneity of functional importance has been described for type 17 cells in mouse models of disease[21,22]. We therefore next investigated molecular heterogeneity within the CCR6+(high)CCR2+ subgroup by single-cell RNA-seq of freshly isolated T cells from three donors after

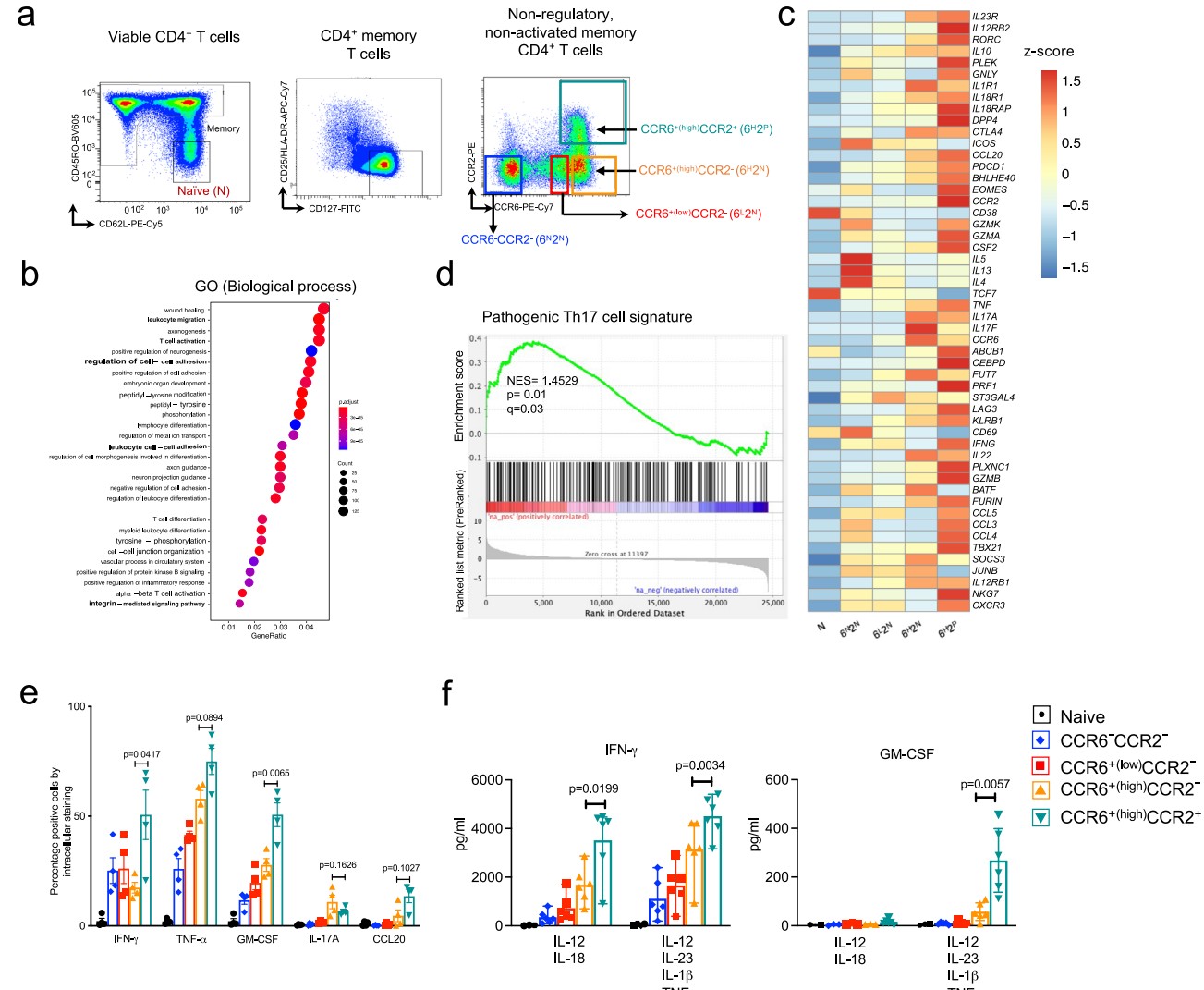

**Fig. 1 | CCR2 identifies type 17 cells with a pathogenic profile. a** Pseudocolor density dot plots showing FACS gating strategy to purify naïve, CCR6⁻CCR2⁻ ($6^N2^N$), CCR6^low^CCR2⁻ ($6^L2^N$), CCR6^high^CCR2⁻ ($6^H2^N$), and CCR6^+(high)^CCR2⁺ ($6^H2^P$) subgroups from CD4⁺ T cells from the blood of healthy human donors. **b** Gene Ontology (GO) biological process gene set analysis of differentially expressed genes ($p < 0.05$) in CCR6^+(high)^CCR2⁺ versus CCR6⁻CCR2⁻ cells. The sizes of the dots represent the numbers of genes in the list of differentially expressed genes associated with the GO terms and the colors of the dots represent the base 2 $p$ values from a one-sided test adjusted for multiple comparisons. **c** Heatmap from bulk RNA-seq showing the expression levels of genes (rows) associated with the pathogenic Th17 phenotype in activated cells in the purified CD4⁺ T cell subgroups as indicated. Color coding reflects standardized gene expression values (z-scores). **d** Gene Set Enrichment Analysis (GSEA) for genes differentially expressed between activated CCR6^+(high)^CCR2⁺ versus CCR6⁻CCR2⁻ cells using a molecular signature gene set for pathogenic Th17 cells and a one-sided test. Normalized enrichment score (NES), nominal $p$ value, and false discovery rate, $q$, which considers multiple comparisons, are shown. **e** FACS-purified cells were activated ex-vivo and stained for the indicated cytokines/chemokine. Gating strategy is shown in Supplementary Fig. 1c. Each symbol shows data from one of $n = 4$ individual donors, analyzed in four separate experiments, and bars indicate means ± SEM. $p$ values were calculated using two-tailed paired Student's $t$ tests, and no corrections were made for multiple comparisons. **f** FACS-purified cells were stimulated ex-vivo with the indicated cytokines and culture supernatants collected after three days were assayed for GM-CSF and IFN-γ by ELISA. Each symbol shows data from one of $n = 6$ individual donors, analyzed from six separate experiments, and bars indicate means ± SEM. $p$ values were calculated using two-tailed paired Student's $t$ tests, and no corrections were made for multiple comparisons. Source data are provided in the Source Data file.

activation with PMA and ionomycin (Fig. 2 and Supplementary Fig. 3). Uniform manifold approximation and projection (UMAP) plots revealed four clusters of cells (Fig. 2a and Supplementary Fig. 3a, d). Cluster dot plots and feature scatter plots (Fig. 2c, d and Supplementary Fig. 3b, c, e, f) showed *CSF2* was expressed broadly among the clusters, whereas most cells with highest expression of the type 17 genes *IL17A* and *IL22* were distinct from those with highest expression of the type 1 genes *IFNG*, *CCL3*, and *CCL4* as well as genes important for cytotoxicity, such as *PRF1*, *GZMK*, and *GZMB*. Additional genes of interest include *IL1R*, which clustered with type 17 cytokines; *EOMES*, which clustered with type 1 cytokines and chemokines; and *IL23R* and *TBX21*, which were broadly distributed.

## Analysis of single CD4⁺ T cells in the CSF of multiple sclerosis patients reveals CCR6⁺CCR2⁺ cells with increased markers of pathogenicity and activation

We next investigated whether CCR6⁺CCR2⁺ Th cells with pathogenic potential analogous to those we defined in the blood of healthy humans might be found at sites of pathogenic tissue inflammation. For this, we used Cellular Indexing of Transcriptomes and Epitopes by Sequencing (CITE-Seq) to analyze cells from the cerebrospinal fluid (CSF) of study participants with multiple sclerosis (MS) and non-MS control participants, since a mouse model of MS, EAE, is the paradigmatic model in which pathogenic type 17 cells have been characterized[19,21,22] - and similar cells have been implicated in the

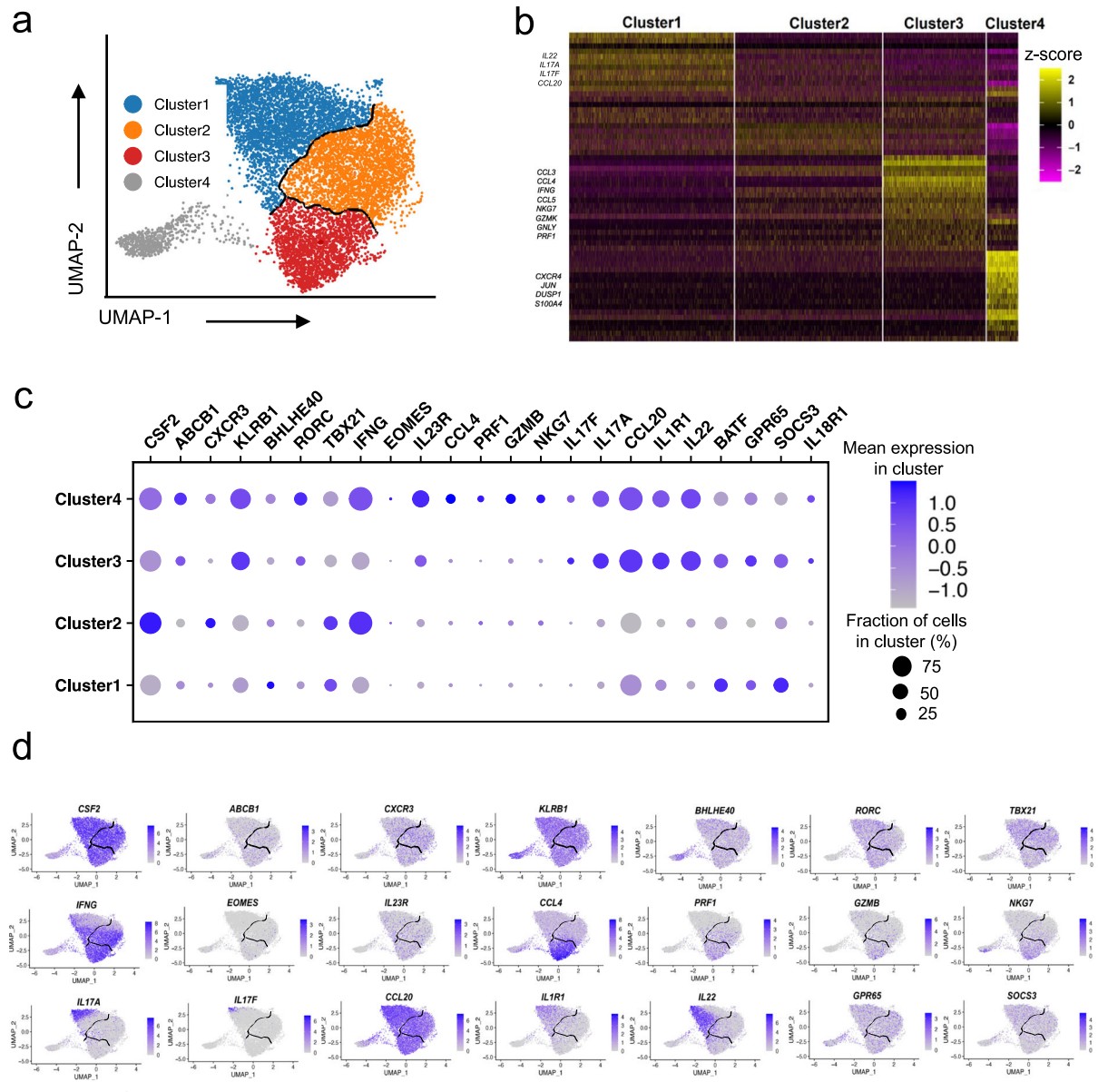

**Fig. 2 | CCR6$^{+(high)}$CCR2$^+$CD4$^+$ T cells separate into clusters expressing *IL17A/F/IL22* vs. *IFNG*. a** Uniform manifold approximation and projection (UMAP) and clustering from scRNA-seq profiles of 8907 CCR6$^{+(high)}$CCR2$^+$ cells from human blood isolated by FACS as in Supplementary Fig. 1a and Fig. 1a from Donor 1 where each dot represents a single cell, with color codes of clusters defining cells with similar transcriptional profiles. **b** Heatmap showing the relative expression across all cells for upregulated and downregulated ($p < 0.05$) genes in each cluster compared to all other clusters. Rows represent genes and columns represent cells grouped by cluster, and some marker genes for clusters 1, 3, and 4 are listed on the left. Color coding reflects standardized gene expression values (z-scores). **c** Dot plot displaying gene expression for a subset of the genes displayed in (**b**), where the size of a dot indicates the percentage of cells in a cluster expressing the indicated gene and the shading represents the average expression level of the gene in the clusters' cells. **d** Feature plots of expression levels of selected genes in cells in the UMAP scatter plot, where the shading represents gene expression levels. Cluster borders are marked in black.

pathogenesis of MS[23,24], including in a study showing pathogenic type 17 cells decreased in the blood and accumulating in the CSF and brain of patients with active disease[25]. Our participants included 19 with MS: eight with relapsing-remitting MS, five with secondary progressive MS, and six with primary progressive MS. There were 10 males and 9 females, ranging from 21 to 68 years of age (median 58). (The sample from a 20th patient with MS was omitted from the final analysis because of treatment with natalizumab, which has a major effect on T cell trafficking to the CNS[26].) The duration of disease ranged from 0.85 to 48.4 years (median 18.1), and the patients' Expanded Disability Status Scale scores ranged from 2 to 7 (median 5). Eight control participants (2 males and 6 females, ranging

from 23 to 69 years of age, median 53) included three individuals with non-inflammatory neurological diseases, one with a non-MS inflammatory neurological disease, and four healthy donors. Samples from 11 MS and three control participants (Batch 1) were analyzed using the antibodies in Supplementary Data 1, excluding the antibody against CCR2, and for samples for eight MS and five control participants (Batch 2) the antibody against CCR2 was included. Five of the MS patients were untreated, and the remaining patients were being treated with B-cell depleting antibodies or immunomodulators, which may have diminished differences between the MS and control samples. Additional information on participants and samples can be found in Supplementary Data 2.

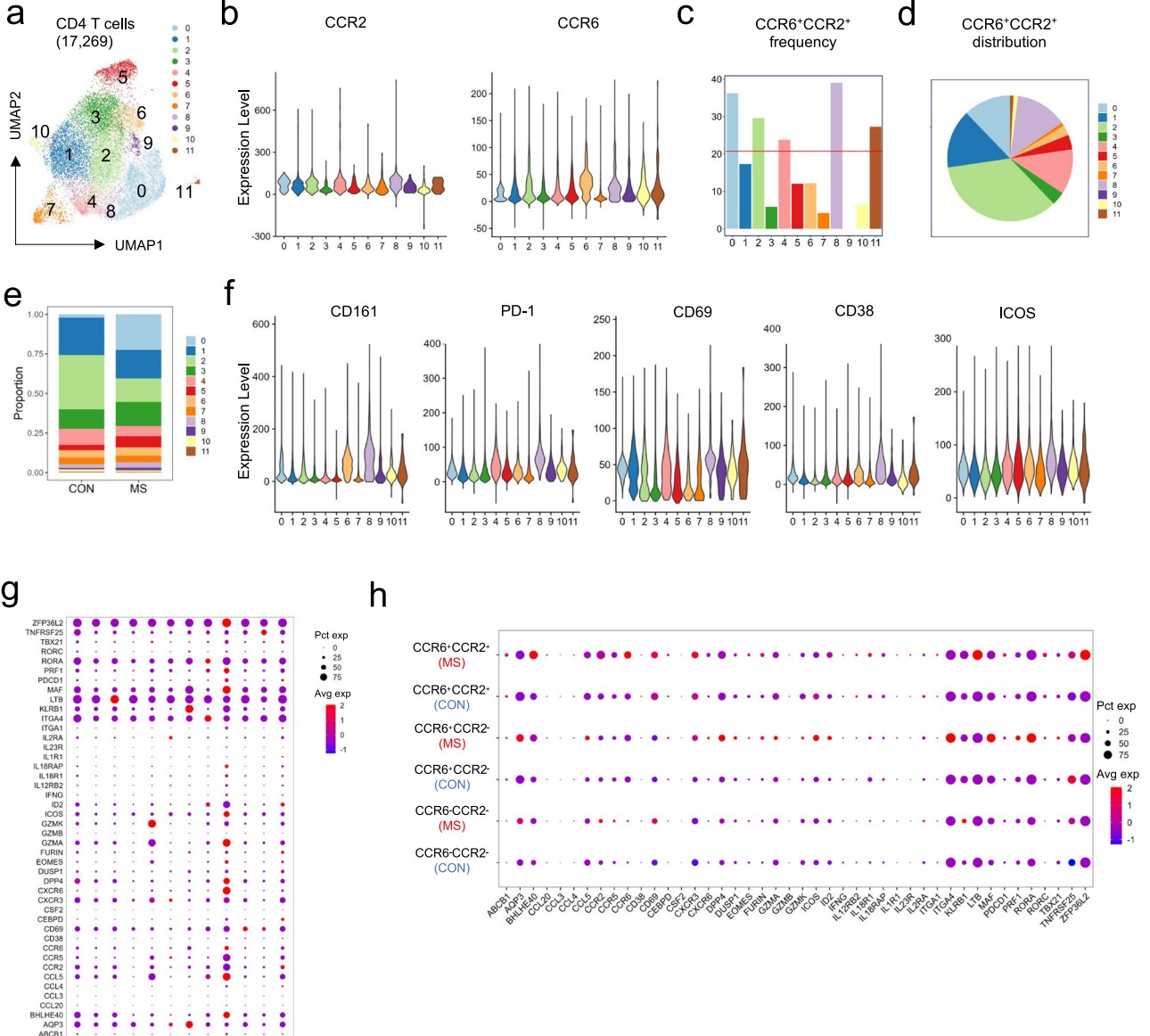

**Fig. 3 | CITE-seq analysis shows pathogenic and activated features for CCR6⁺CCR2⁺CD4⁺ T cells in the CSF of patients with multiple sclerosis. a** UMAP and clustering of single cells based on RNA expression data from CITE-Seq of cells isolated from the cerebrospinal fluid of eight healthy controls and 19 MS patients (Batches 1 and 2, see text). Each dot represents an individual cell, color-coded according to its cluster identity. **b** Violin plots illustrating the expression of surface CCR6 and CCR2 across the UMAP clusters using antibody-derived data from Batches 1 and 2. **c** Bar graph showing percentages of CCR2⁺CCR6⁺CD4⁺ T cells in each cluster, using antibody-derived data from Batch 2. The red, horizontal line indicates the percentage of all cells that are CCR6⁺CCR2⁺. **d** Pie chart showing the proportions of the CCR6⁺CCR2⁺ cells that are found in each cluster, using antibody-derived data from Batch 2. **e** Proportions of UMAP clusters (**a**) in cells from control and MS participants. **f** Violin plots illustrating the expression of surface CD161, PD1, CD69, CD38 and ICOS across the UMAP clusters in (**a**), using antibody-derived data from Batches 1 and 2. **g** Dot plot visualizing the expression of key pathogenic Th17-associated genes across UMAP clusters in (**a**) where the size of a dot indicates the percentage of cells in a cluster expressing the indicated gene and the shading represents the average expression level of the gene in the clusters' cells. **h** Dot plot visualizing the expression of key pathogenic Th17-associated genes for the cells from MS versus control (CON) participants analyzed according to surface expression of CCR6 and CCR2, using antibody-derived data from Batch 2, where dots represent the data as in (**g**).

Most of the T cells in the CSF of both MS and control participants are CD4⁺[27]. UMAP plots generated from CD4⁺ T cells in our samples yielded 11 clusters (Fig. 3a). Of the major clusters, the adjacent clusters 0, 2, and 8 had significant numbers of cells expressing both *CCR6* and *CCR2* and showed the greatest enrichment in cells co-expressing surface CCR6 and CCR2 (Fig. 3b, c, see Supplementary Fig. 4a for identification of cells expressing surface CCR6 and CCR2). Cluster 6 was highly enriched in CCR6⁺ cells but had few cells that were CCR2⁺. Together, clusters 0, 2, and 8 contained a majority (~61%) of the cells co-expressing surface CCR6 and CCR2 (Fig. 3d). Clusters 0 and 8 (but not 2) were relatively expanded within the CD4⁺ T cell population in

the CSF from the patients with MS as compared to the control samples (Fig. 3e).

In our analyses of the CITE-Seq data we expanded the list of genes/proteins of interest to include those with increased expression in cells implicated in pathogenicity in MS[28] and those characteristic of T cells that are resident in the human CNS[26,29]. The pattern of enrichment in surface protein and gene expression for clusters 0 and 8 (Fig. 3f, g)—and particularly for cluster 8 - matched the pattern identified in pathogenic cells found in the CNS in mice with EAE[19,22] and/or in cells proposed to be pathogenic in MS[28]. These genes/proteins include CCR2 itself and *CXCR6*, PD-1(*PDCD1*), *TBX21, BHLHE40, IFNG*,

CD161(*KLRB1*), *MAF*, and *RORA*, as well as a collection of CD8[+] T cell-associated genes, including *PRF1*, granzyme genes, and *CCL5*[19,22,30–32]; and cluster 8 showed enrichment for markers of T cell activation, including CD69 (also high in cluster 0) and CD38 as well as ICOS, which has a role in Th17 cell effector function and in EAE[33,34].

Resident memory CD4[+] T cells in human brain have been characterized as having mixed Th17/Th1 features[26,29], and cluster 8 also showed relatively high expression of markers of these cells, some of which overlap with markers of pathogenicity, and include *CCR5*, *CXCR6*, PD-1(*PDCD1*), *GZMA*, *ZFP36L2*, and *ID2*. By contrast, cluster 0 does not show relatively high expression of resident markers but does show higher expression of some genes associated with pathogenic cells based on studies of MS patients, including *ITGA4* and *TNFRSF25*[28]. Cluster 6 was enriched in CCR6[+]CCR2[-] cells, but except for CD161(*KLRB1*), which is highly expressed on Th17 cells, and *AQP3*[28], this cluster did not show notable expression of genes associated with pathogenicity or activation in MS/EAE (Fig. 3g). In analyzing clusters 0, 2, 6, and 8 in samples from patients with MS versus control (non-MS) samples for expression of these genes, the most notable differences were for cluster 8, which showed, in the patient samples, increases in percentages of expressing cells and/or levels of expression for, among others, *BHLHE40*, *CCL5*, *ICOS*, *PDCD1*, *LTB*, *PRF1*, and *MAF*, a pattern suggesting cells with enhanced pathogenicity and activation (Supplementary Fig. 4b).

We next analyzed patterns of gene expression in the cell subgroups defined using the surface expression of CCR6 and CCR2 (regardless of cluster designation), analogous to the way we characterized cells from the blood of healthy donors and compared the data from cells from MS patients versus cells from control participants (Fig. 3h). The frequency of CCR6[+]CCR2[+] cells was not higher in the samples from the patients versus the controls (Supplementary Fig. 4c). However, as compared with the CCR6[+]CCR2[+] subgroup from the control samples, the CCR6[+]CCR2[+] subgroup in the MS samples showed increases in percentages of expressing cells and/or levels of expression for genes associated with a pathogenic phenotype in mouse and/or human cells, including *ABCB1*, *BHLHE40*, *CCR6* and *CCR2*, *CCR5*, *CD69*, *CXCR6*, *KLRB1*, *LTB*, and *PDCD1*[20,22,28,30,32,34,35], plus relatively high expression of the activation marker, *CD69*. A few of these genes and others expressed in pathogenic and activated cells were also increased in the CCR6[+]CCR2[-], and, to a lesser extent, in the CCR6[-]CCR2[-] subgroup in cells from the patients versus controls, including *AQP3*, *BHLHE40*, *ICOS*, *ITGA4*, *KLRB1*, *MAF*, and *RORA*. Taken together, these data show that the CD4[+]CCR6[+]CCR2[+] T cells in the CSF cluster disproportionately with cells with resident memory and pathogenic features, and that the pathogenic features, together with markers of activation, are enhanced in this cell subgroup in the CSF of patients with MS versus control participants. Differences in gene expression at the level of cell clusters or subgroups in the CSF of patients with MS versus control participants could represent gene induction/repression and/or changes in cell composition due, for example, to the recruitment of new cells.

## CD4[+]CCR6[+(high)]CCR2[+] T cells are the cells that migrate most efficiently across monolayers of cytokine-activated endothelial cells

We next used flow chambers with cytokine-activated human umbilical vein endothelial cells (HUVECs) to study the trafficking behavior of the CCR6[+(high)]CCR2[+] cells as compared with the other CD4[+] subgroups. We set shear stress at 0.75 dyne/cm$^2$ for 4 min to allow for T cell accumulation, after which the shear stress was increased to 3 dyne/cm$^2$ for an additional 16 min. Data were collected by video microscopy for subsequent analysis and are displayed as cells per field that demonstrated the sequential steps of rolling, firm arrest, and TEM. Given the requirements for rolling before arrest and arrest before TEM, any change in the number of cells undergoing an earlier step will affect the number of cells undergoing a subsequent step. Therefore, to evaluate the steps of arrest and TEM independently we also display, as percentages, ratios of numbers of arrested/rolling cells and transmigrating/arrested cells.

As compared with the four subgroups of memory cells, fewer naïve cells were able to roll and virtually none was able to arrest and undergo TEM. Within the memory population there was a general increase in numbers of cells rolling and arresting and in the percentages of rolling cells arresting from the CCR6[-]CCR2[-] to CCR6[low]CCR2[-] to CCR6[+(high)]CCR2[-] to CCR6[+(high)]CCR2[+] cells, and only the CCR6[+(high)]CCR2[+] cells were able to undergo TEM (Fig. 4a).

Because chemokine receptor activation contributes variably to T cell arrest[36], we assessed trafficking in the system in the presence and absence of pertussis toxin, which inhibits $G_{i/o}$ proteins, the principal mediators of chemokine receptor signals. Pertussis toxin had no effect on rolling, diminished arrest by 50%-90% depending on the cell subgroup, and completely blocked TEM of the CCR6[+(high)]CCR2[+] cells (Fig. 4b). To address the roles of individual chemokine receptors in these processes, we blocked specific receptors that we selected based both on the frequencies of their expression on the memory cell subgroups and the levels of induction of chemokine-encoding mRNAs in the TNF-α-activated HUVECs (Supplementary Fig. 5a). Blocking CCR6 activity using a neutralizing antibody against CCL20 led to a significant reduction in arrest specifically of CCR6-expressing cells but had no effect on the percentage of arresting CCR6[+(high)]CCR2[+] cells undergoing TEM (Fig. 4 c). Blocking CCR5 with an antagonist, maraviroc, resulted in a modest reduction in arrest for the CCR6[+(high)]CCR2[-] and CCR6[+(high)]CCR2[+] cells (Fig. 4d). In the top panel the effect of maraviroc on diminishing arrest for the CCR6[+(high)]CCR2[-] cells did not quite reach statistical significance ($p = 0.063$), whereas the $p$ value was 0.0312 when samples were analyzed specifically for the arrest step using the arrested/rolling cell ratio (bottom panel). Maraviroc inhibition of TEM for the CCR6[+(high)]CCR2[+] cells (Fig. 4d, top panel) appears to be explained by its inhibition of arrest, since blocking CCR5 had no consistent effect on TEM of these cells as reflected in the transmigrating/arrested cell ratios (Fig. 4d, bottom panel). By contrast, inhibiting CCR2 using the antagonist BMS22 had no effect on arrest but almost eliminated TEM by the CCR6[+]CCR2[+] cells (Fig. 4e), demonstrating a dedicated role for CCR2 in this final step.

To enhance the inflammatory stimuli in the flow chamber assay and expand the collection of chemokines and receptors available to contribute to arrest and TEM, we activated HUVECs with both TNF-α and IFN-γ. The addition of IFN-γ led to a general increase in arrest of the memory cell subgroups, with no effect on rolling or on the percentages of arrested cells undergoing TEM (Fig. 5a). Combining these two cytokines increased expression of the mRNAs for the CCR5 ligand CCL5 and two CXCR3 ligands, CXCL10 and CXCL11, and induced expression of the CXCR3 ligand, CXCL9, and CCR2 ligands CCL7 and CCL8 as compared with untreated cells (Supplementary Fig. 5b). Blocking CXCR3 with an antagonist, AMG 487, mainly diminished arrest, particularly on the CCR6[-]CCR2[-] and CCR6[low]CCR2[-] cells (Fig. 5b), which expressed little to no CCR6 but high levels of CXCR3 (Supplementary Fig. 2b). Taken together, our experiments using HUVECs in flow chambers show redundant but context-dependent activities of CCR6, CCR5, and CXCR3 in memory T cells that are limited to arrest and a non-redundant role for CCR2 in TEM.

## The restricted role for CCR2 in TEM is consistent with ligand localization

To understand the basis for the specialized activities of chemokine receptors with activated HUVECs, we next investigated the localization of the HUVEC chemokines. We first measured chemokine found in the culture medium (supernatant) versus cell-associated chemokine for CCL20 and CCL2 after treating HUVECs with TNF-α ± IFN-γ or for CCL7 and CCL8 after treating HUVECs with TNF-α + IFN-γ. In the activated

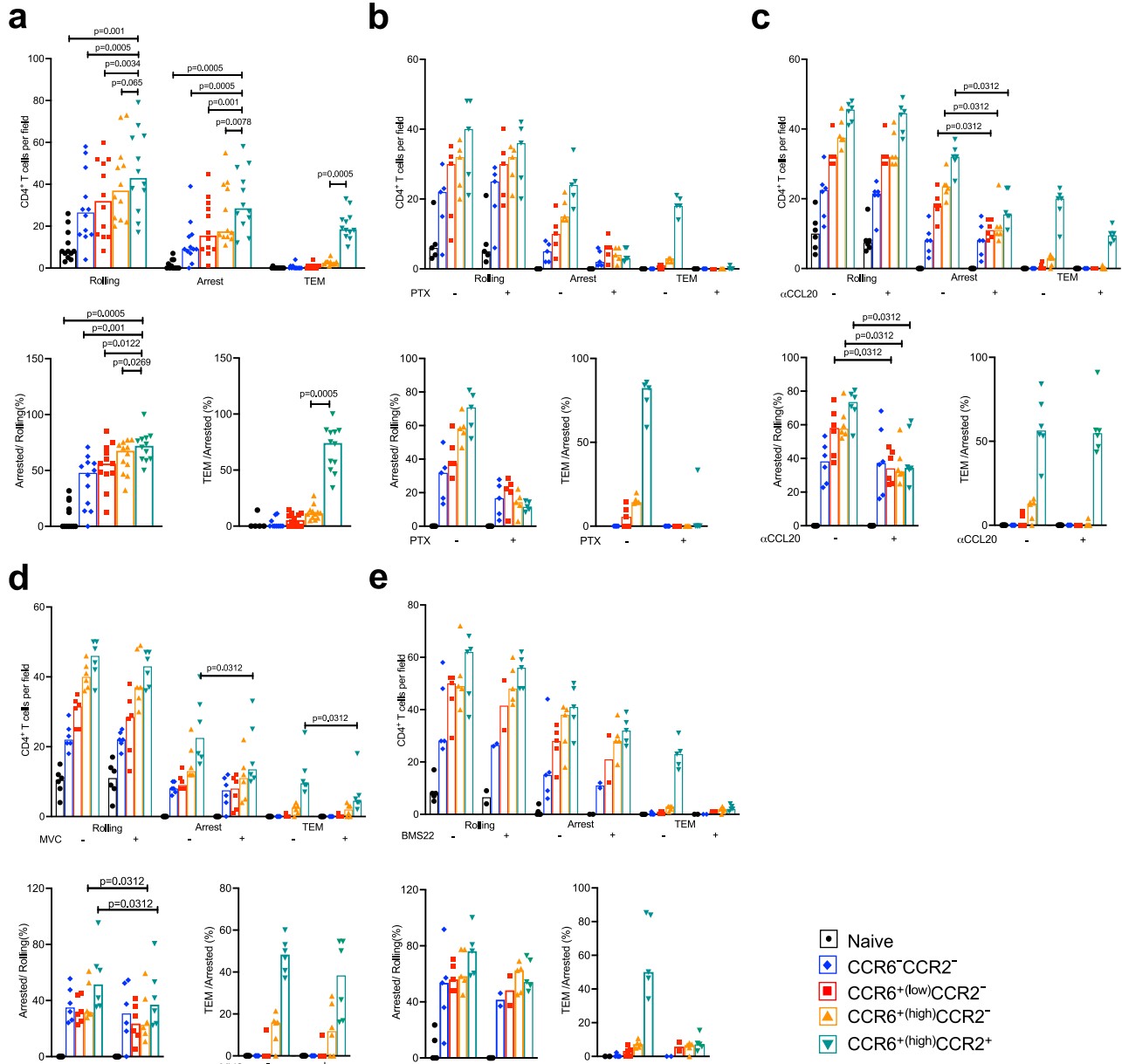

**Fig. 4 | Unlike other subgroups of CD4$^+$ T cells, CCR6$^{+(high)}$CCR2$^+$ cells are able to undergo TEM in flow chamber assays. a** Numbers of cells rolling, arrested and transmigrated on TNF-$\alpha$-activated HUVECs for human CD4$^+$ T cell subgroups isolated by FACS from the blood of healthy donors as in Supplementary Fig. 1a and Fig. 1a. Lower panels show arrested cells as percentages of cells rolling and transmigrated cells as percentages of cells arresting. Each symbol shows data from one of $n = 12$ individual donors, analyzed from 12 separate experiments, and bars indicate medians. **b** Data for T cells either untreated (marked with a minus sign) or treated with pertussis toxin (PTX, $n = 5$ individual donors and experiments). Data are displayed as in (**a**). Data from the five untreated samples are also shown in (**a**). **c** Data for T cells in flow chambers using HUVECs pretreated with IgG control (marked with a minus sign) or anti-CCL20 antibody ($\alpha$CCL20, $n = 6$ individual donors and experiments). Data are displayed as in (**a**). Data from one untreated

sample are also shown in (**a**, **b**). **d** Data for T cells either untreated (marked with a minus sign) or treated with the CCR5 antagonist, maraviroc (MVC, $n = 6$ individual donors and experiments). Data are displayed as in (**a**). **e** Data for T cells either untreated (marked with a minus sign) or treated with the CCR2 antagonist, BMS22 ($n = 5$ individual donors and experiments). In two of the five experiments all the T cell subgroups were treated with BMS22, whereas in the three additional experiments only the CCR6$^{high}$CCR2$^-$ and CCR6$^{+(high)}$CCR2$^+$ cells were treated with BMS22. Data are displayed as in (**a**). Data from two untreated samples are also shown in (**a**). $p$ values were calculated using a two-tailed Wilcoxon matched-pairs signed rank test, and no corrections were made for multiple comparisons. Source data are provided in the Source Data file and representative videos used to quantify rolling, arrest, and transendothelial migration are provided in Supplementary Movies 1–15.

HUVECs, CCL20 was mainly cell-associated whereas all three CCR2 ligands tested were found primarily in the supernatants (Supplementary Fig. 5c). We next tested whether CCR2 ligand secretion by HUVECs was polarized by culturing the cells on transwell membranes with or without TNF-$\alpha$ and IFN-$\gamma$ and measuring chemokine content in the upper and lower wells. All three CCR2 ligands were secreted into both the upper and lower wells, with some bias for the lower wells (Fig. 6a).

To localize chemokines within and on ECs, we used antibodies to stain HUVECs treated with TNF-$\alpha$ and IFN-$\gamma$ both before and after cell permeabilization. Ligands for CCR6 (CCL20), CCR5 (CCL5), and CXCR3 (CXCL9) were detected both on the cell surface and within HUVECs by confocal microscopy, while the ligand for CCR2 (CCL2) was only detected intracellularly (Fig. 6b). Similarly, the other CCR2 ligands, CCL7 and CCL8, were found to be secreted but were not or minimally

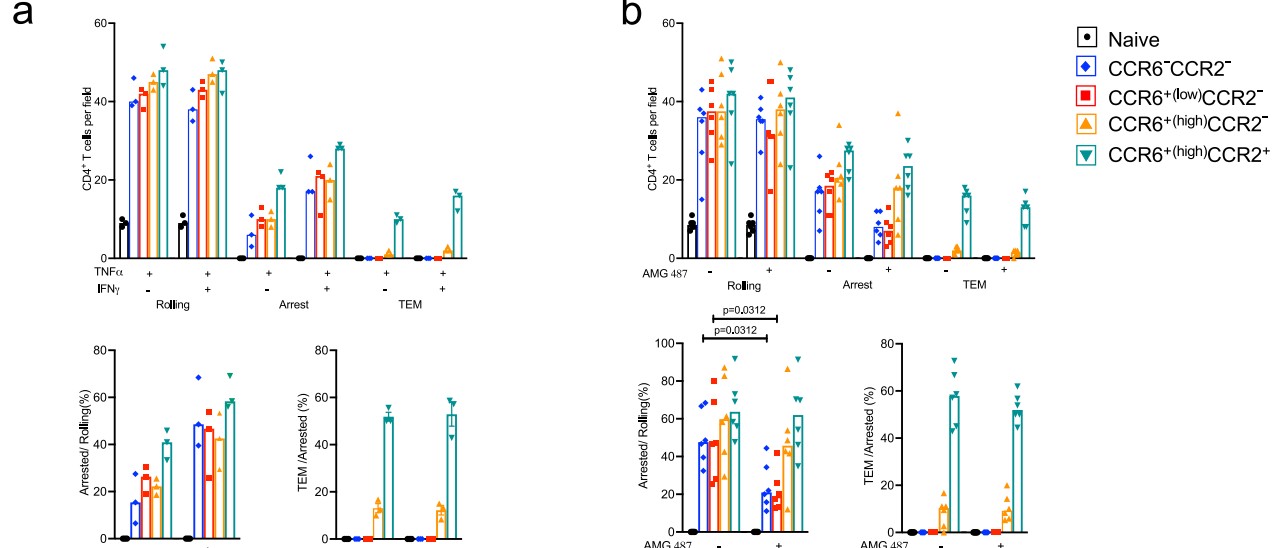

**Fig. 5 | CXCR3 mediates arrest of CD4⁺ T cells on TNF-α + IFN-γ-activated HUVECs.** Human CD4⁺ T cell subgroups were isolated from the blood of healthy donors by FACS as in Supplementary Fig. 1b and Fig. 1a. **a** Numbers of cells rolling, arrested and transmigrated on HUVECs activated with TNF-α alone or TNF-α + IFN-γ. Lower panels show arrested cells as percentages of cells rolling and transmigrated cells as percentages of cells arresting. Each symbol shows data from one of n = 3 individual donors, analyzed from three separate experiments, and bars indicate medians. **b** Data for T cells either untreated (marked with a minus sign) or

treated with the CXCR3 antagonist, AMG 487 (n = 6 individual donors and experiments) using HUVECs treated with TNF-α + IFN-γ. Data are displayed as in (**a**). Three samples with T cells not treated with AMG 487 were also used in the experiments in (**a**) comparing HUVECs treated with TNF-α alone versus TNF-α + IFN-γ. p values were calculated using two-tailed Wilcoxon matched-pairs signed rank tests, and no corrections were made for multiple comparisons. Source data are provided in the Source Data file, and representative videos used to quantify rolling, arrest and transendothelial migration are provided in Supplementary Movies 16–23.

detected on the cell surface (Fig. 6b). Importantly, no signal was observed when using the secondary antibody alone (see Supplementary Fig. 8b), confirming the specificity of the staining.

Additionally, we tested the binding of CCL2 and CCL5 to untreated and TNF-α-activated HUVECs using exogenous synthetic proteins that were biotinylated to facilitate selective detection on the cell surface using streptavidin-phycoerythrin, avoiding a background signal from the corresponding endogenously produced protein. Consistent with our results for endogenous chemokines, synthesized CCL5, but not CCL2, effectively bound to the surface of HUVECs. No binding was observed with streptavidin-PE alone, nor on fibronectin-coated plates without cells, confirming the specificity of CCL5 binding. These results were similar regardless of whether HUVECs were left untreated or activated with TNF-α (Supplementary Fig. 5d).

To determine whether this differential pattern of chemokine localization is specific to HUVECs, we tested human dermal microvascular endothelial cells (HDMECs) treated with TNF-α and IFN-γ. As for HUVECs, all three CCR2 ligands were detected almost exclusively within the cells, while the ligands for CCR6 (CCL20), CCR5 (CCL5), and CXCR3 (CXCL9) were present both on the cell surface and within the HDMECs (Supplementary Fig. 6a). Likewise, in flow chamber assays, although the CCR6⁺(high)CCR2⁻ and CCR6⁺(high)CCR2⁺ T cells both rolled and arrested efficiently on HDMECs, only CCR6⁺(high)CCR2⁺ cells underwent TEM. Moreover, blocking CCR2 with the antagonist BMS22 had no effect on arrest of either subgroup of cells on HDMECs but significantly impaired TEM of the CCR6⁺(high)CCR2⁺ cells (Supplementary Fig. 6b).

The binding of chemokines to cell surfaces and extracellular matrix is by interactions with glycosaminoglycans (GAGs) such as heparan sulfate (HS) and chondroitin sulfate, and we presumed that the differences we found in chemokine binding to ECs were due to differences in GAG binding. We tested this hypothesis using parental CHO-K1 cells, which express GAGs including HS, the major GAG on CHO cells[37], and CHO D-677 cells, which lack HS. We tested HS-

dependent binding to these cells for CCL5, and CCL2, again using synthesized chemokines with site-specific biotinylation, which we detected using streptavidin-PE and flow cytometry. CCL5 showed binding to CHO-K1 cells that was partially abrogated using the CHO D-677 cells, demonstrating binding to HS. By contrast, CCL2 showed no HS-dependent binding (Fig. 6c). Because heparan sulfate is a critical GAG for chemokine binding to ECs[38], the differential binding of these chemokines to cell-surface HS likely contributes to the differences we found in their binding to HUVECs and HDMECs.

In our staining experiments, chemokines on EC surfaces appeared in micro-aggregates rather than being distributed homogeneously. This has been observed previously[39,40], and although surface binding patterns can be influenced by similar inhomogeneities in the distribution of HS, there is not a simple correspondence between chemokine and HS distributions[39]. In the permeabilized ECs, we found that the most intense staining for CCL2, CCL8 and CXCL9 was perinuclear in an asymmetric pattern suggesting accumulation in the Golgi apparatus (Fig. 6b), which was verified by co-staining these cells for GOLPH2 (Supplementary Fig. 7). Because some chemokines are stored in Weibel-Palade bodies[41], we co-stained permeabilized HUVECs for chemokines and von Willebrand factor (vWF). None of the chemokines we analyzed here co-localized with vWF, which was found around the nucleus in a symmetrical pattern (Supplementary Fig. 8a), consistent with results reported previously for CCL2 and CCL5[41].

## CCL2 bound to HUVECs mediates CCR2-dependent arrest and inhibits TEM

Our data on the localization of chemokines suggested that the cause of CCR2's failure to contribute to T cell arrest was a deficiency of CCR2 ligand on the surface of ECs. Nonetheless, CCR2 may still have been activated by surface ligand below the limit of detection of our immunofluorescent staining assay, or even by low levels of secreted ligands in the flowing medium, and CCR2's failure to mediate arrest may, instead, have been due to an intrinsic property of CCR2 rather than an

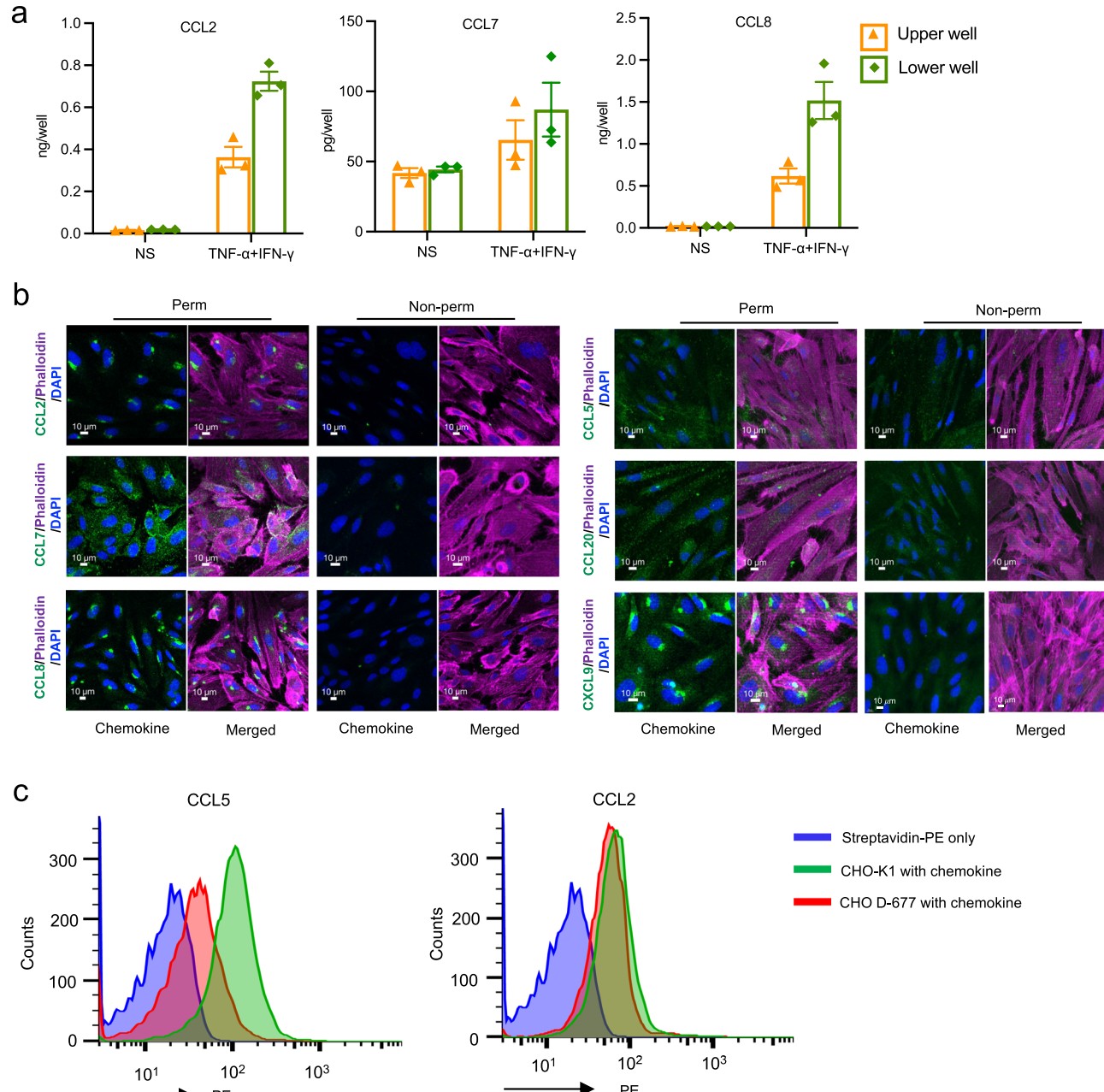

**Fig. 6 | CCR2 ligands are secreted but do not bind to HUVECs. a** Culture medium collected from the upper and lower wells of transwells with HUVEC monolayers either not stimulated (NS) or stimulated with TNF-α + IFN-γ were assayed for CCL2, CCL7, and CCL8 by ELISA ($n = 3$ separate experiments). Each symbol shows data from one experiment, with assays done in duplicate, and bars indicate means ± SEM. **b** Confocal microscopy images at × 40 magnification of permeabilized and non-permeabilized TNF-α + IFN-γ-stimulated HUVECs immunostained for CCL2, CCL7, CCL8, CCL5, CCL20, and CXCL9 (green) and stained using phalloidin for polymerized actin (magenta) and DAPI for nuclei (blue). The scale bars indicate 10 μm. Images are representative of three experiments. **c** Biotinylated CCL5 (0.1 μM) and CCL2 (1 μM) were incubated with CHO-K1 cells and the heparan-sulfate deficient cell line, D-677. Chemokine binding was detected using streptavidin-phycoerythrin (PE) and flow cytometry. Data shown are representative of three separate experiments.

insufficiency of ligand. In one approach to address this possibility, we tested CCR2's capabilities by adding CCL2 to CCR6⁺(high)CCR2⁺ cells immediately before loading them into the flow chamber and maintained the CCL2 in the medium for the first four min of the assay when cells are allowed to roll and arrest under low shear stress. Under these conditions, CCL2 did, in fact, increase the cells' arrest, and this increase was reversed by blocking CCR2. It was of interest that while adding CCL2 enhanced arrest, TEM was suppressed (Fig. 7a). Although inhibition of TEM could have been due to receptor internalization, pre-incubating cells with CCL2 at the concentration used in the flow chamber assay (100 ng/ml) did not lead to loss of surface CCR2

(Supplementary Fig. 9). Nonetheless, this concentration of CCL2 presumably resulted in desensitization of surface receptors (without internalization) that persisted long enough to prevent CCR2-mediated signaling and TEM during the remaining 16 min of the assay.

In a second approach, rather than using soluble ligand we sought to produce a more physiologically appropriate CCR2 arrest-mediating ligand by having ECs secrete a form of CCL2 that would bind to surface GAGs, analogous to ligands for other chemokine receptors. This was done by transducing HUVECs with a lentiviral vector containing sequences encoding a chimeric chemokine consisting of full-length human CCL2 followed by carboxy-terminal residues 74-103 of the

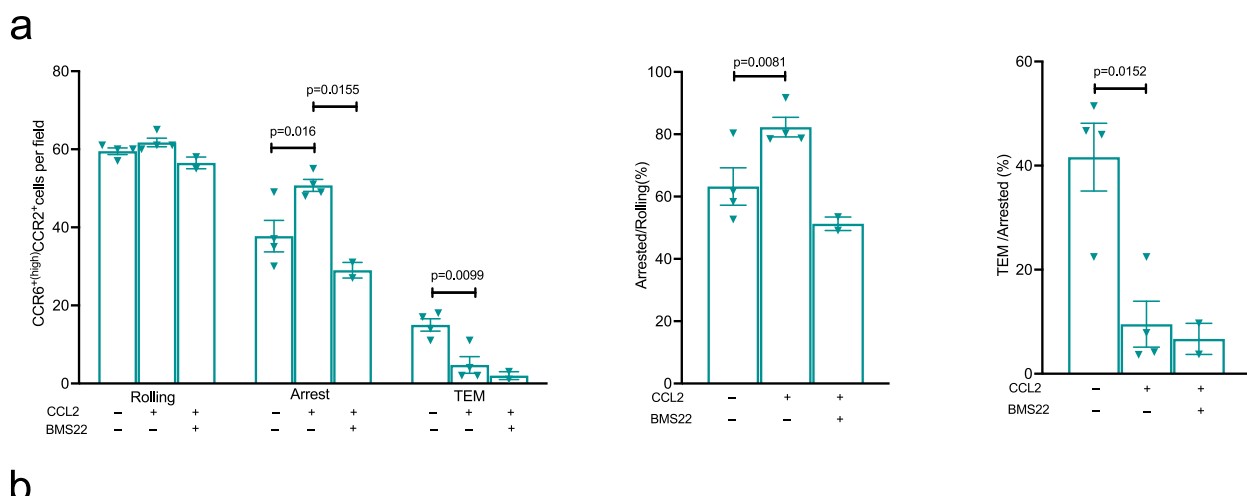

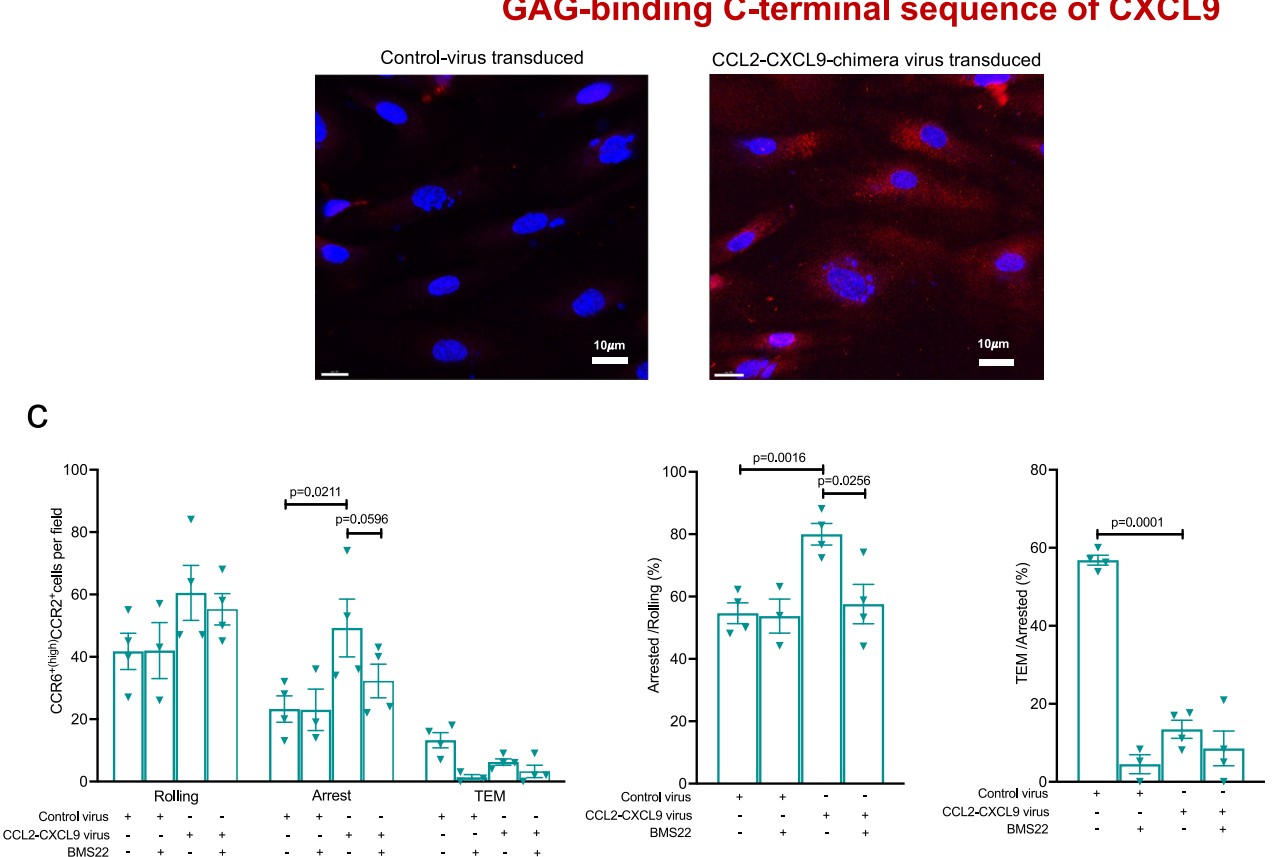

**b**

**CCL2-CXCL9 Chimera**

MKVSAALLCLLLIAATFIPQGLAQPDAINAPVTCCYNFTNRKISVQRLASYRRITSSKCPKEAVI
FKTIVAKEICADPKQKWVQDSMDHLDKQTQTPKTKKKQKNGKKHQKKKVLKVRKSQRSRQKKTT

**GAG-binding C-terminal sequence of CXCL9**

Control-virus transduced

CCL2-CXCL9-chimera virus transduced

secreted, human CXCL9. These carboxy-terminal sequences of CXCL9 are highly basic[42,43] and have been shown to bind GAGs[43], and we demonstrated that HUVECs transduced with the chimeric sequences now showed surface staining with the antibody directed against CCL2 (Fig. 7b). We purified control-transduced and chimera-transduced GFP[+] HUVECs by FACS, and after activating them with TNF-α used them in flow chamber assays with CCR6[+(high)]CCR2[+] cells. We found increased numbers of CCR6[+(high)]CCR2[+] cells arresting on the chimera-expressing HUVECs, and these increases were reversed by inhibiting CCR2. Additionally, the surface-bound CCL2-CXCL9 chimera effectively blocked TEM (Fig. 7c). Taken together, these data suggest that an insufficient concentration of ligand resulting from an inability of ligand to bind to the surface of ECs under conditions of flow, rather than any intrinsic property, determined CCR2's failure to induce arrest of the CCR6[+(high)]CCR2[+] cells. Further, the inhibition of TEM by CCL2 presented on the luminal side suggests TEM requires that T cells sense a transendothelial chemokine gradient.

## Discussion

From our previous work characterizing the phenotypes of human effector/memory Th cells based on their pattern of chemokine receptor expression[16] and roles of specific chemokine receptors in the

**Fig. 7 | Surface-bound CCL2 can induce CCR2-mediated arrest and suppress TEM.** For (**a**, **c**), human CD4+ T cell subgroups were isolated by FACS from the blood of healthy donors as in Supplementary Fig. 1a and Fig. 1a.
**a** CCR6+(high)CCR2+ cells were either left untreated (marked with a minus sign) or treated with CCL2 (100 ng/ml) just before and for four min after adding the cells to the flow chambers, and CCL2-treated cells were used without (marked with a minus sign) or with treatment with the CCR2 antagonist, BMS22. Data points in the left panel show numbers of cells rolling, arrested, and transmigrated, while the middle panel shows arrested cells as percentages of cells rolling and the right panel shows transmigrated cells as percentages of cells arresting. Each symbol shows data from one donor, with n = 4 individual donors in four separate experiments using CCL2. Two of these experiments included the additional treatment with BMS22. **b** Above is the CCL2-CXCL9 chimera sequence with signal peptide sequence shown in green, CCL2-sequence in black, and C-terminal GAG-binding sequence of CXCL9 in red. Below are confocal microscopy images of control or CCL2-CXCL9-chimera

transduced, TNF-α-stimulated HUVECs immunostained for CCL2 (red) and stained with DAPI for nuclei (blue). Images are representative of three experiments.
**c** Numbers of CCR6+(high)CCR2+ cells, either untreated (marked with a minus sign) or treated with the CCR2 antagonist, BMS22, rolling, arrested, and transmigrated on TNF-α-activated HUVECs transduced with either control virus (marked with a minus sign) or virus encoding the CCL2-CXCL9 chimera. Middle panel shows arrested cells as percentages of cells rolling and in the right panel transmigrated cells as percentages of cells arresting. Each symbol shows data from one donor, with cells from four donors and separate experiments in each treatment group except for the BMS22-treated cells with control virus-transduced HUVECs, where cells from three donors were used. Bars indicate means ± SEM. p values were calculated using two-tailed paired Student's t tests, and no corrections were made for multiple comparisons. Source data are provided in the Source Data file, and representative videos used to quantify rolling, arrest, and transendothelial migration are provided in Supplementary Movies 24–27.

steps of extravasation of human CD8α+ MAIT cells[18], we reasoned that CCR2 might mark a pro-inflammatory subgroup of CCR6+/type 17 cells that have, as one of their functionalities, the ability to migrate efficiently into inflamed tissue. We found that the profile of gene and protein expression of CCR6+(high)CCR2+ Th cells in the blood resembles what has been reported for pathogenic Th17 cells, and described, according to the experimental context, in mice or humans as ex-Th17, Th1-like, Th1*, or Th17.1 cells[8,25,44,45].

In mouse models these cells mediate pathology in EAE[21], autoimmune colitis[44] and arthritis[46]. In humans, these cells have been implicated in host defense against mycobacteria[47] and in the autoimmune diseases of MS[23–25] and rheumatoid arthritis[48]. To manifest pathogenicity in mice these cells have been reported to require IL-23[19,49] and the transcriptional regulators RORγt and T-bet[31,44], BHLHE40[30], and EOMES[50]. The critical effector cytokines that have been shown to mediate these cells' pathogenic activity have included GM-CSF[31,46,49–53], IFN-γ[31,44], and IL-17[52]. Some of these cells express a cytotoxic program[22], including multiple granzymes that may be important for tissue damage[32]. Surface proteins described on these cells in humans include CCR6 and CXCR3[8,45,48] and MDR1/ABCB1[20,48]. We found that many of the relevant genes, including *IL23R, IL18R1, IL18RAP, RORC, TBX21, BHLHE40, EOMES, CSF2, IFNG, CXCR3, PRF1, GZMB, GZMK, GNLY, NKG7,* and *ABCB1* were enriched in the CCR6+(high)CCR2+ vs. the CCR6+CCR2- subgroups of human Th cells from the blood and the percentages of cells capable of producing GM-CSF and IFN-γ after pharmacologic activation ex vivo were similarly increased in the CCR6+(high)CCR2+ vs. the CCR6+CCR2- subgroups.

The contributions of CCR6+(high)CCR2+ cells to immunopathology, as well as to host defense, particularly early during pathogen challenge, would be enhanced if these cells were able to produce effector cytokines in response to an inflammatory environment in the absence of cognate antigens. We found that CCR6+(high)CCR2+ cells were the CD4+ T cells that secreted the most IFN-γ and GM-CSF solely in response to combinations of inflammation-associated cytokines. Recent data in mice have suggested that this sort of cytokine-induced, bystander activation of CD4+ T cells can, in fact, contribute to the immunopathology in EAE[54].

Clustering of CCR6+(high)CCR2+ cells from our single-cell RNA-seq analysis of cells from the blood revealed differences in the distributions of mRNAs encoding proteins associated with pathogenicity. Of particular interest, within the major clusters, there was widespread expression of *CSF2*, whereas overlapping expression of *IL17A/F* and *IL22* with *IFNG* was limited. The distribution of expression of *IL1R1* matched that of *IL17A/F*, while expression of *CCL4, PRF1,* and *EOMES* matched that of *IFNG*. The coincidence of expression of *IL1R1* and *IL17A/F* is consistent with the critical role for IL-1β in the differentiation of human Th17 cells, in marking Th17 cells and enhancing their production of IL-17A and, together with IL-12, in enabling human CCR6+ Th cells to acquire a pathological phenotype[8,55]. In the UMAP-generated

display, co-localization of expression of *IFNG* with the chemokine genes *CCL3, 4,* and *5* is consistent with the established co-expression of these chemokines with IFN-γ[56]. The expression of *PRF1*, encoding perforin, and/or genes encoding granzymes has been reported in Th cells, including Th17 cells[57], that are implicated in autoimmunity[58] and host defense[57] - and that co-express EOMES and IFN-γ[58]. Together, our data suggest that although expression of *CSF2* is a general property of the CCR6+(high)CCR2+ Th cells, these cells tend to fall along a gradient between two phenotypes: (1) of cells expressing *IL1R1* and able to induce *IL17A/F* and *IL22*, and (2) of cells with type 1 features, including the ability to induce *IFNG* and genes for several CC chemokines, as well as expression of *EOMES* and genes important for cytotoxicity.

In mice, a recent publication has identified CD4+CCR6+CCR2+ cells as GM-CSF producers contributing to the immunopathology in dextran sodium sulfate-induced colitis[59], and CCR2 itself has been shown to be required for EAE-mediated disease due to the receptor's critical role in monocyte migration[60]. CCR2 has also been reported to contribute to EAE by mediating recruitment to the central nervous system of a GM-CSF-producing subgroup of cells characterized as CCR6−CCR2+[53]; and CCR2+CCR5+ Th cells that could produce IFN-γ in response to myelin basic protein were found to be enriched in the CSF of patients with MS during relapse[35].

In analyzing CITE-seq data of CD4+ T cells from the CSF of patients with MS and non-MS control participants, we found that pathogenicity-associated genes identified in EAE in mice, such as *BHLHE40, TBX21,* and *IFNG*, as well as a collection of CD8+ T cell-associated genes, including *PRF1*, granzyme genes, and *CCL5*[19,22,30–32], were differentially expressed in CCR6+CCR2+ cell-enriched UMAP clusters, some of which were expanded in the CSF of patients with MS. In addition, these cell clusters in the CSF showed comparatively increased expression of other genes/proteins associated with the encephalitogenic phenotype, including PD-1(*PDCD1*), *CCL5*, and *CXCR6*[19,22] and for genes/proteins expressed by cells proposed to be pathogenic in MS, such as CD161(*KLRB1*), *CCR5, ITGA4, TNFRSF25, RORA,* and *MAF*[28,35]. It is notable that some of the preferentially expressed genes in one of the CCR6+CCR2+-cell enriched clusters, cluster 8, also characterize the CCR5hi TH17.1 cells described as resident T cells in the brain/CSF[26,29], which have a phenotype that overlaps the one described for pathogenic cells and which have been shown to be particularly efficient at trafficking across brain ECs[29]. In analyzing CSF cells defined solely by surface expression of CCR6 and CCR2, we found increases in percentages of expressing cells and/or levels of expression for a subset of pathogenicity-associated genes, including *ABCB1, BHLHE40, CCR6* and *CCR2, CCR5, CXCR6, KLRB1, LTB,* and *PDCD1*, along with relatively high expression of the activation marker, CD69, in the CCR6+CCR2+ subgroup from the MS patients versus the non-MS control participants. These differences could reflect gene induction/repression and/or changes in cell composition due, for example, to the recruitment of new cells. Taken together, our data suggest that

CCR6⁺CCR2⁺ cells are type 17 cells with a pathogenic signature that, although resting in the blood, can enter the CNS and, in patients with MS, show increases in markers of pathogenicity and activation that suggest the potential to contribute to disease-associated inflammation.

The ability to migrate into inflamed tissue is integral to the function of effector-capable T cells, which can serve as early responders from the blood to reinforce resident memory cells during pathogen challenge or to contribute to immunopathology[6]. Although there have been a few highly informative studies of roles for individual chemokine receptors on human T cells[61,62], the current study describes, to our knowledge, the only investigations of roles for individual chemokine receptors on non-manipulated populations of human effector/memory Th cells in the steps important for extravasation. Our studies provide evidence that chemokine receptors may subserve sequential specialized roles on type 17 cells with pathogenic potential, expanding our understanding of the multistep paradigm for T cell diapedesis. Using activated, primary human ECs in flow chamber assays, we have shown here the activities of CCR5, CCR6, and CXCR3 in firm arrest, but no convincing role for these receptors in TEM. By contrast with these receptors, CCR2 was unable to contribute to arrest, but was the only receptor we found consistently to be a mediator of TEM. Chemokine ligands for the arrest-mediating receptors could be detected on the surface of the ECs. By contrast, although secreted by both adherent and non-adherent sides of the activated HUVECs, ligands for CCR2 could hardly be detected on the EC surfaces.

The display of chemokines on the surfaces of ECs depends on binding to GAGs, primarily heparan sulfate (HS)[39,63], and eliminating HS diminishes leukocyte trafficking to lymph nodes[64] and inflamed tissue[38]. Although the complexity and heterogeneity of GAGs has hindered the identification of the physiological moieties binding to chemokines, existing data suggest significant specificity in GAG-chemokine interactions, including for CCL5, CCL2, and CCL20[63,65]. Although CCL5 has a particularly high affinity for GAGs[65], CCL2 and CCL7 also bind[63,66], and CCL2 binding to HUVEC[65] has been reported. Nonetheless, other investigators have also described poor/undetectable binding of CCL2 to ECs[36,67–69], including to HUVECs, HDMECs, or human ECs from saphenous vein, lung, or brain[69].

Neither published data nor our data explain why it is difficult to detect binding of CCL2 (or CCL7 or CCL8) to HS/GAGs on cell surfaces even though these chemokines can be demonstrated to bind heparin, HS, and other GAGs in simplified assays. Modifications of or complexes formed by endogenous CCL2, or competition for GAG binding by other chemokines secreted by activated ECs would seem unlikely explanations given our and/or others' difficulties in detecting the binding even of synthesized or recombinant CCL2 to the surfaces of either non-activated or activated HUVECs. We found that although CCL5 showed HS-dependent binding to CHO cells, CCL2 did not, suggesting that poor binding to HUVECs and HDMECs could result from general difficulty in CCL2's binding to cell-surface HS, perhaps reflecting differences in composition between the cell-surface HS and the reagents used in typical binding studies of chemokines and GAGs. Nonetheless, our data do not preclude the possibility that CCL2 or other CCR2 ligands may bind to the surfaces of some ECs in vivo, such as has been reported for HEVs in mouse lymph nodes[64] and human dermal venules[70]. The latter experiments used ¹²⁵I-labeled CCL2 and CCL7[70], reinforcing the obvious point that detection depends on the sensitivity of the assay, and whereas our findings using immunofluorescence show little evidence of binding of CCR2 ligands to ECs or of biological activity of these ligands on the EC surface, we cannot rule out that surface binding of these chemokines could be detected using more sensitive techniques.

CCR2-mediated arrest after adding CCL2 to T cells immediately before and during the initial, low stress portion of the flow chamber assay, or, more interestingly, by using HUVECs displaying GAG-bound CCL2 as part of a CCL2-CXCL9 chimera, demonstrated that the failure of CCR2 to mediate arrest was solely due to the lack of sufficient ligand at the site and time of the initial T cell-HUVEC interactions. We showed that CCL2 and other CCR2 ligands are secreted from both sides of HUVECs. We presume that dilution of unbound chemokine due to continuous flow on the luminal side coupled with basilar secretion establishes a transendothelial concentration difference that drives CCR2-dependent TEM. The inhibition of TEM from exposing cells to high concentrations of surface bound CCL2 on the luminal side would follow from the loss of this transendothelial difference and/or an inability to respond to a chemokine gradient due to receptor desensitization.

It has been reported that the delivery of CCL2 to invasive filopodia at sites of T cell-endothelial cell contact at the luminal surface by intracellular vesicles is a mechanism for CCR2-dependent TEM of activated T cells[36]. However, other studies support the importance of a transendothelial difference in ligand concentration for CCR2-mediated TEM of leukocytes[61,67,68]. Additionally, there are data suggesting the preferential secretion of chemokines from the basal surface of ECs[61], which would be expected to enhance a transendothelial difference. Figure 8 shows a cartoon illustrating our view of the steps and mediators of trafficking of the CCR6⁺(high)CCR2⁺ cells.

Our model suggesting differing roles for chemokine receptors in lymphocyte arrest versus TEM depending on whether chemokines are or are not EC-bound raises several questions. Firstly, although it is clear from experiments using Boyden chamber assays under static conditions that leukocytes can migrate in response to a transendothelial difference in CCL2 concentration[67], precisely how a leukocyte senses a chemokine gradient across the endothelial cell barrier remains uncertain. Consistent with our findings, elegant studies of the extravasation of mouse neutrophils in vivo have shown separate roles for chemoattractants and/or chemoattractant receptors in arrest versus TEM[71,72]. In these reports, the CXCR2 ligand(s) mediating TEM were produced by non-endothelial cells and transported and/or presented on ECs by the atypical chemokine receptor, ACKR1, which was shown to localize preferentially at endothelial cell junctions[71]. In our experiments, the activated ECs themselves were the source of the CCR2 ligands mediating TEM, thereby obviating a requirement for ACKR1 in endothelial cell transport and/or presentation, and we found no evidence of CCR2 ligands localized at endothelial cell junctions. Nonetheless, it is possible that chemokine secreted on the abluminal side could be presented at the junctions by endothelial cell ACKR1 and form a gradient sensed by the leukocytes' invading podosomes[73]. In this regard, both CCL2 and CCL7 bind well to ACKR1[74]. Alternatively, leukocytes could sense soluble gradients extending from the abluminal side into the endothelial cell junctions, which are opened at the luminal side by leukocyte-endothelial cell interactions prior to the initiation of TEM[75].

Secondly, the inhibition of TEM by the CCL2-CXCL9 chimera might suggest that T cell migration should likewise be inhibited by endothelial-cell bound chemokines such as CCL5, CCL20, and CXCL9, since moving away from the luminal surface could involve migrating in opposition to the concentration difference in chemokine. However, based on our and published data[61], like CCL2, other chemokines, including arrest-mediating chemokines that bind to ECs, are also secreted at both luminal and basal sides of ECs, minimizing the size of any transendothelial gradient that would oppose TEM. Additionally, in vitro experiments on neutrophil migration have shown both that cells can migrate up a concentration gradient of one chemoattractant in the presence of a saturating concentration of a second chemoattractant and that there are hierarchies of responses such that cells can migrate up the gradients of some chemoattractants even while moving down the gradients of others[76]. One or both processes could be operating in the sequential responses that we found to chemokines mediating arrest followed by TEM. Further, it is increasingly

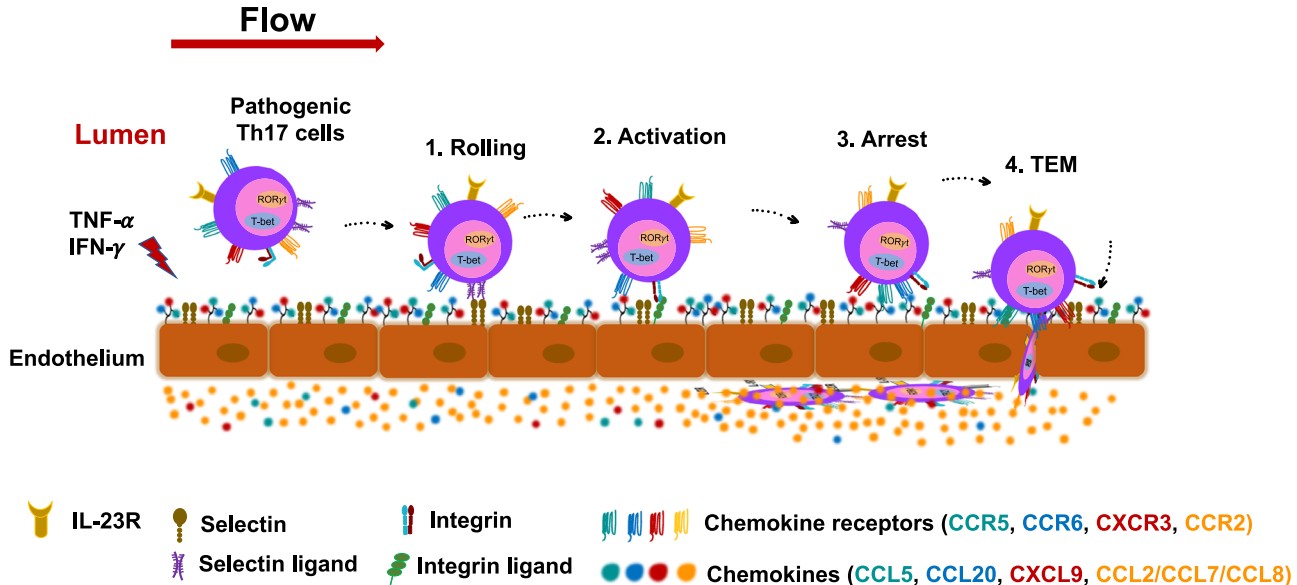

**Fig. 8 | Trafficking of pathogenic-like human type 17 Th cells across activated endothelial cells.** Cytokine-activated endothelial cells upregulate expression of selectins and integrin ligands and secrete chemokines. Selectin-selectin ligand interactions help capture cells and allow them to roll along the endothelium. For pathogenic-like type 17 cells, endothelial cell-bound chemokines can then stimulate CCR6, CCR5, and CXCR3 to activate integrins for binding to intercellular adhesion molecules and mediate firm arrest. Secreted chemokines that fail to adhere to endothelial cells, such as CCL2 and other CCR2 ligands, are removed by blood flow to create a transendothelial gradient that drives TEM. Although for the sake of clarity the T cell depicted here co-expresses the four chemokine receptors, not all CCR6+(high)CCR2+CD4+ T cells express CCR5 and/or CXCR3.

appreciated that chemokine-receptor mediated chemokine internalization by migrating leukocytes consumes chemokine, making these cells chemokine sinks that can shape gradients for chemotaxis[77–79]. This process could also diminish retention signals by chemokines bound on the luminal surface to facilitate responses to transendothelial gradients, serving as one of many mechanisms that leukocytes could use to navigate environments with competing chemoattractants[80].

Next, there is the question of the generalizability of our findings. CCR2 is very unlikely to be the only chemokine receptor able to mediate TEM of memory T cells into inflammatory sites. We presume that differences among tissue ECs, GAG expression, and chemokine repertoire could allow ligands and their receptors to exchange roles in mediating arrest versus TEM during extravasation. How universal is the separation in the activities of arrest versus TEM among chemokine receptors? There are examples where a single receptor appears to be mediating both steps in extravasation. One such case is CCR7-driven migration of naïve T cells across HEVs into lymphoid organs – and the HEV is an example of a tissue-specific venule containing ECs with a unique chemokine profile, namely in their constitutive production of CCL21 during homeostasis[81]. However, studies of CCR7 on dendritic cells have revealed a separation of the activities of firm adhesion versus gradient-driven migration mediated, respectively by full-length CCL21, which contains GAG-binding residues at its C-terminus, and a C-terminal-truncated CCL21 or CCL19, which do not bind GAGs and can form soluble gradients[82]. In contradistinction to CCL19[83], CCL21 is the critical CCR7 ligand for T cells to enter lymph nodes through HEVs during homeostasis. Since both the full-length and truncated CCL21 are found in lymph nodes[82], these two forms may be functioning for extravasation across HEVs analogously to the surface-bound and non-bound chemokines, respectively, in our flow chamber assays with HUVECs and HDMECs. An important difference, however, between what we observed in treating CCR6+(high)CCR2+ T cells with CCL2 and the published findings for CCR7+ dendritic cells, was that for CCR7 only surface-bound chemokine was able to produce integrin-dependent adhesion[82], whereas we were able to enhance T cell arrest on activated

HUVECs by pre-treatment with soluble CCL2. We can speculate that whereas CCR7 produces different signals from surface-bound versus non-bound ligands, allowing one receptor to mediate both adhesion and chemotaxis, CCR2, and perhaps the other receptors on T cells mediating trafficking at inflammatory sites lack this capability, requiring that these functions be served by separate receptors.

Finally, there are conditions under which, in contrast to our data and the case of CCR7 and its ligands, it appears that a single form of a single ligand acting through a single chemoattractant/chemokine receptor can mediate both adhesion/arrest and TEM. In one study, gels impregnated with CXCL2 placed on the cremasteric muscles in mice induced asymmetric adhesion of neutrophils on venules and biased extravasation toward the CXCL2 gel[84]. This study also showed that binding of CXCL2 on EC HS was necessary for neutrophil arrest, and that HS was required to create an intraluminal gradient of CXCL2 that mediated directional extravasation. A second study showed that superfusion of joints in mice with several single chemoattractants, including CXCL1 and CXCL2, could induce both neutrophil arrest and TEM[72]; and other studies showed that single chemokines expressed at tissue sites from transgenes can induce the accumulation of mononuclear cells and the formation of tertiary lymphoid structures[85,86].

Because chemokines can function as secretagogues, it is possible that in these models one chemokine can induce the secretion of additional chemoattractants that cooperate to stimulate extravasation. Assuming this is not the case, and one chemokine-receptor pair can, in fact, mediate extravasation, we can speculate that the process would require not only chemokine binding to GAGs on the endothelium to trigger arrest but also a transendothelial gradient of the proper magnitude to accomplish TEM in the face of persistent luminal, surface-bound chemokine. Difficulty in meeting these requirements to achieve efficient and site-specific extravasation may explain why in more complex in vivo models and environments, in contrast to the artificial conditions in the experiments described above, neutrophils use different chemoattractant receptors and/or different ligands (which may be producing different signals) for arrest versus TEM[71,72] - and dendritic cells use different CCR7 ligands that separately mediate

adhesion versus chemotaxis[82]. The CCR6[+(high)]CCR2[+] T cells we studied in the current report offer a robust solution by using differentially localizing chemokines and their separate receptors to mediate these distinct steps of extravasation.

Although our work has been limited to analyses of human samples and experiments done ex vivo, the strength of our studies is the integrated investigation of multiple aspects of the effector activities of bona fide human type 17 Th cells. We have identified a strong connection between the expression of CCR2 on CCR6[+]/type 17 cells and a pattern of gene and protein expression associated with pathogenicity in immune-mediated disease and host defense against mycobacteria, and have found that CD4[+]CCR6[+]CCR2[+] T cells express markers of pathogenicity and activation in the CSF of patients with MS. We have investigated in detail how multiple chemokine receptors, expressed in complex and combinatorial patterns on human memory Th cells, are coordinated for accomplishing the steps required for extravasation through the differential localization of their ligands, with several receptors mediating arrest, but with CCR2 being essential for migration across monolayers of cytokine-activated ECs - a critical component of type 17 effector function. These studies have revealed fundamental principles of chemokine system functioning on lymphocytes, raising key questions for future research, and establishing a more informed basis for considering interventions to block or enhance T cell trafficking into tissue.

## Methods

### Donors and participants
Human elutriated lymphocytes from healthy donors were collected by the Department of Transfusion Medicine, Clinical Center, National Institutes of Health, under a protocol approved by the Institutional Review Board, protocol number 99-CC-0168 https://clinicalstudies.info.nih.gov/protocoldetails.aspx?id=99-CC-0168&&query=, which is listed under NCT00001846 at https://clinicaltrials.gov. Cerebrospinal fluid was collected from 19 participants with MS and eight control participants under a protocol approved by the Institutional Review Board of the National Institutes of Health, protocol number 09-I-0032 https://clinicalstudies.info.nih.gov/protocoldetails.aspx?id=09-I-0032&&query=09-I-0032, which is listed under NCT00794352 at https://clinicaltrials.gov. For both protocols, all participants provided written, informed consent, but did not provide consent for release of personally identifiable information. Participants in protocol 99-CC-0168 were compensated, which was also true for some but not all participants in protocol 09-I-0032. For protocol 09-I-0032, sex was determined by self-report and was not considered in the study design.

### Cell culture
Cryopreserved primary HUVECs (Cat# PCS-100-013), CHO-K1 (Cat# CCL-61) cells and pgsD-677 (CHO D-677, Cat# CRL-2244) cells were purchased from ATCC, and HDMECs were purchased from LONZA (Cat# CC-2543). Both HUVECs and HDMECs were cultured in complete vascular cell growth medium (Vascular Cell Basal Medium: (Cat# PCS-100-030), supplemented with Endothelial Cell Growth Kit-VEGF (Cat# PCS-100-041; ATCC) to yield final concentrations of rh VEGF: 5 ng/mL, rh EGF: 5 ng/mL, rh FGF basic: 5 ng/mL, rh IGF-1: 15 ng/mL, L-glutamine: 10 mM, heparin: 0.75 units/mL, hydrocortisone: 1 μg/mL, ascorbic acid: 50 μg/mL, fetal bovine serum: 2%, also containing penicillin-streptomycin (100 U/mL, Cat# 15140122; Gibco) and maintained at 37 °C in 5% CO$_2$. To subculture, HUVECs were washed once with PBS, treated with 0.05% trypsin-EDTA (Cat# 25300054; Thermo Fisher Scientific) for 90 s and were used until passage three. After purification by FACS (see below), human primary CD4[+] T cells were cultured overnight in RPMI-10 medium (RPMI 1640, Cat# 11875119; Gibco), penicillin-streptomycin (100 U/ml, Cat# 15140122; Gibco) and HEPES (10 mM, Cat# 15630080; Gibco) containing 10% FBS (Cat# 100-500; Gemini Bio-product) at 37 °C in 5% CO$_2$ before using in

experiments. CHO-K1 and D-677 cells were cultured in F12K medium (30-2004, ATCC) supplemented with 10% fetal bovine serum.

### Purification of T-cell subgroups from blood, cell sorting, and culturing with and measuring cytokines
Human CD4[+] T cells were isolated from ~2 × 10[8] elutriated lymphocytes to ~90% purity by negative selection using RosetteSep human CD4[+] T cell enrichment cocktail (Cat# 15062, Stem Cell Technologies), using Ficoll/Hypaque (Cat# 45-001-749, Cytiva) density centrifugation. If needed, red blood cells were removed using ACK lysing buffer (Cat# BP10-548E, Lonza). Cells were incubated with anti-human CCR2-biotin (Cat# FAB151B, R&D Systems) in FACS buffer (HBSS, Cat# 14025092; Gibco) plus 4% fetal bovine serum for 30 min at room temperature (RT) and following washing, cells were stained with streptavidin-PE (Cat# 405245), anti–human CD4-Brilliant Violet 421 (Cat# 317434), anti–human CD45RO-Brilliant Violet 605 (Cat# 304238), anti-human CD62L-PE-Cy5 (Cat# 304808), anti-human CD25-APCCy7 (Cat# 302614), anti-human HLA-DR-APCCy7 (Cat# 307618) anti-human CD127 FITC (Cat# 351314), all from BioLegend and anti-human CCR6 PECy7 (Cat# 560620; BD Biosciences) for 30 min at RT. Cell subgroups were sorted to nearly 100% purity using FACSAria II Cell Sorter (BD Biosciences). For treatment with cytokines, cells were purified as above, except the anti-human CCR2 antibody was PE-conjugated (Cat# 357206, BioLegend), and cultured in IL-12 (20 ng/mL, Cat# 219-IL-005; R&D Systems), IL-18 (20 ng/mL Cat# 9124-IL-010; R&D Systems), IL-23 (20 ng/mL, Cat# 1290-IL-010/CF; R&D Systems), IL-1β (20 ng/mL Cat# 201-LB-005/CF; R&D Systems) and TNF-α (20 ng/mL Cat# 210-TA-005/CF; R&D Systems) in medium containing 10 ng/mL IL-7 (Cat# BT-007-010; R&D Systems) at 37 °C in 5% CO$_2$ and after 3.5 days, supernatants were collected and concentrations of IFN-γ and GM-CSF were determined using ELISA kits from R&D Systems (IFN-γ; Cat# DIF50C, GM-CSF; Cat# DGM00) according to the manufacturer's instructions.

### Bulk RNA-seq
T cells were cultured unstimulated or stimulated with cell stimulation cocktail (00-4970-03; Invitrogen) for 3 h at 37 °C in 5% CO$_2$. Extraction of total RNA was done from all samples using RNeasy mini kit (Qiagen) according to the manufacturer's instructions. Oligo dT-based mRNA enrichment was done using NEBNext Poly(A) mRNA Magnetic Isolation Module (Cat# E7490L) and library preparation was completed using NEBNext Ultra II Directional RNA Library Prep Kit for Illumina (Cat# E7760L), with 50 ng total RNA input per sample and targeting for 200 bp RNA insert size and NEBNext Multiplex Oligos for Illumina (Unique Dual Index Primers, Cat# E6440L). Sequencing was done using an Illumina NextSeq 500 system. The resulting FastQ files were checked for quality using FastQC (Babraham Bioinformatics tool). Adapter trimming was done using trimmomatic (v0.32[87],); alignment to the human reference genome was done with STAR[88]; duplicate reads were removed using Picard MarkDuplicates tool; and number of reads per gene per sample were counted using the featureCounts software from the subread package[89]. Differential gene expression analysis was done in R (v4.2.1) using DESeq2 package (v1.38.1)[90] and results were filtered for genes with <5% probability of being false positive (adjusted $P < 0.05$). GO analysis was done in R using clusterProfiler[91]. Gene Set Enrichment Analysis (GSEA)[92,93] was done using GSEA software.

### Intracellular staining for flow cytometry
T cells purified by FACS were stimulated with Leukocyte Activation Cocktail, with GolgiPlus (Cat# 550583; BD Pharmingen) for 6 h at 37 °C with 5% CO$_2$ before being stained with following antibodies alone or in combinations: anti-IL-17A (Cat# 512305), anti-IFN-γ (Cat# 502530)), anti-TNF-α (Cat# 502944), anti-GM-CSF (Cat# 502305) all from BioLegend and anti-CCL20 (Cat# C360A; R&D Systems) by using Cytofix/CytoPerm Plus kit (Cat# 555028; BD Pharmingen). Samples were

analyzed using an LSRFortessa flow cytometer (BD Biosciences), and the data were analyzed using FlowJo software (BD Biosciences).

### Single-cell RNA sequencing (scRNA-seq)

CD4$^+$CCR6$^+$CCR2$^+$ and total unselected memory T cells were stimulated with cell stimulation cocktail (Cat# 00-4970-03; Invitrogen) for 3 h at 37 °C in 5% CO$_2$. Cells were loaded onto the Chromium Single Cell Library & Gel Bead Kit v3.1 (10x Genomics), to generate libraries for scRNA-seq and sequenced using the HiSeq 4000 System (Illumina). Each library was down sampled to equivalent sequencing depth and libraries were merged with Cell Ranger aggr v2.0.2 pipeline (10X Genomics). Data were imported into R Studio (R 3.6.2), and the Seurat package (Seurat 3.1.4, https://github.com/satijalab/seurat) was used to process and analyze the gene expression data. After quality control, genes expressed in fewer than 10 cells were filtered out, and cells with more than 500 genes detected and fewer than 5% mitochondrial genes were retained for analysis. Data were standardized and normalized, and the principal component analysis (PCA) was implemented for nonlinear dimension reduction. Finally, cluster analysis was used to identify cell subtypes and UMAP was used for visualization of dimension reduction results. The FindAllMarkers function in the Seurat package was used to identify the significantly differentially expressed genes (DEGs) of every cluster vs. all others.

### Cerebrospinal fluid collection, processing and analysis for single-cell CITE-seq

Cerebrospinal fluid (CSF) was collected on ice and processed by investigators blinded to the participants' diagnoses, clinical, and imaging data. Aliquots were assigned alphanumeric identifiers and centrifuged at 300 × $g$ for 10 min at 4 °C within 30 min of collection. CSF cells were labeled with 79 TotalSeq B (Biolegend) (see Supplementary Data 1) and washed with PBS containing 1% bovine serum albumin (Thermo Fisher Scientific). Labeled cells were then fixed using the Fixed RNA Feature Barcode Kit (10X Genomics) and stored at −80 °C. CITE-seq was employed to analyze transcriptome and surface protein data from single cells simultaneously. Following probe hybridization, gene expression and surface protein libraries were generated using Chromium Next GEM Single Cell 3' LT Reagent Kits v3.1 (Dual Index Kit, 10X Genomics), according to the manufacturer's protocol (user guide CG000400). The single-cell fixed RNA and feature barcode libraries were sequenced using a NextSeq 2000 platform (Illumina) with a P3 28-cycle + 90-cycle asymmetrical run.

Raw data were processed and quality-checked using Cell Ranger v7.0 (10X Genomics). Doublet cells were predicted and excluded using DoubletFinder v2.0[94], and protein libraries were de-noised with the dsb package[95]. Cells with high mitochondrial gene content (>10%) or excessive unique molecular identifiers (UMIs) (>8000) were removed. Filtered read count data were merged using canonical correlation analysis (Seurat v4, reciprocal PCA method[96],). The merged dataset was clustered using Seurat with a K-nearest neighbor (KNN) graph and UMAP. Non-biased clustering was performed with weighted nearest neighbor analysis, and annotation was based on human peripheral blood mononuclear cell data[96].

### Total RNA isolation and real-time RT-PCR

For detection of chemokine gene expression in HUVECs, cells were cultured as described above, either unstimulated or stimulated with TNF-α and/or IFN-γ for 18 h, and total RNA was isolated using RNeasy mini kit (Qiagen) according to the manufacturer's instructions. Real-time RT-PCR was performed using 50 ng of RNA and the Platinum Quantitative RT-PCR ThermoScript One-Step System with FAM primers and probe sets from Applied Biosystems (*CCL2*, Hs00234140_m1; *CCL5*, Hs00982282_m1; *CCL7*, Hs00171147_m1; *CCL8*, Hs04187715_m1; *CCL20*, Hs00355476_m1; *CXCL9*, Hs00171065_m1; *CXCL10*, Hs00171042_m1;

*CXCL11*, Hs00171138_m1). Values for chemokine mRNAs were normalized based on the values for *GAPDH* (Hs02786624_g1).

### Analysis of CD4$^+$ T cell migration under flow

μ-Slide I$^{0.4}$ Luer parallel plate flow chambers (Cat# 80176; Ibidi) were coated with fibronectin (50 μg/mL, Cat# 1918-FN; R&D Systems) in PBS and HUVECs were plated at confluence and were stimulated with 40 ng/mL recombinant human TNF-α (Cat# 210-TA; R&D Systems) and, in some experiments, 50 ng/mL IFN-γ (Cat# 285-IF; R&D Systems) for 18–20 h at 37 °C in 5% CO$_2$. HUVEC-coated parallel plate flow chambers were assembled with a two-pump system (Harvard Apparatus). T cells were re-suspended at 4 × 10$^5$ cells/mL in RPMI-10 medium. Perfusion of T cells into the flow chambers was performed at 37 °C under a shear stress 0.75 dyne/cm$^2$ for 4 min to allow accumulation of T cells, followed by a constant shear stress of 3 dyne/cm$^2$ for 16 min. Images were acquired at a rate of four frames per second with an integrated fluorescence microscope, Leica AF 6000LX (Leica Microsystems Inc.) with a 20 × DIC objective. Data were analyzed using Imaris (Oxford Instruments) to track rolling, arrest and TEM. We categorized rolling cells as cells with a rolling interaction with the EC monolayer before detaching or arresting; arresting cells as cells that remained stopped on the HUVEC monolayer for more than 10 s under a sheer stress of 3 dyne/cm$^2$; and transmigrating cells as cells that went underneath through stepwise darkening under a sheer stress of 3 dyne/cm$^{97}$. For inhibiting Gi/o proteins, T cells were pre-incubated with 1 μg/mL pertussis toxin (3097; R&D Systems) in RPMI-10 medium for 3 h at 37 °C. For blocking CCR2, cells were pre-incubated for 30 min with BMS CCR2 22 (BMS22, 2 μM, Cat# 3129), for blocking CCR5 with maraviroc (10 μM, Cat# 3756) and for blocking CXCR3 with AMG487 (10 μM, Cat# 4487), all from R&D Systems, at 37 °C, and the inhibitors were left in the medium throughout the assays. For neutralizing the CCR6 ligand, CCL20, HUVECs in flow chambers were pre-treated with 20 μg/ml anti-human CCL20 antibody (Cat# MAB360; R&D Systems) for 2 h at 37 °C, and antibody was maintained at 10 ng/mL throughout the assays. For treatment of T cells with CCL2, recombinant human CCL2 (100 ng/mL, Cat# 279-MC-010; R&D Systems) was added to the cells immediately before loading them into the flow chamber and the CCL2 concentration was maintained during the initial 4 min of the assay.

### Measuring chemokines by ELISA

For detection of chemokine secretion, HUVECs were left unstimulated or stimulated with TNF-α and/or IFN-γ for 18 h in culture conditions. Medium was collected, and cells were lysed in medium containing 0.1% Triton X-100. Concentrations of chemokines were determined using ELISA kits from R&D Systems (CCL2, Cat# DCP00; CCL7, Cat# DCC700; CCL8, Cat# DY281; CCL20, Cat# DM3A00) according to the manufacturer's instructions. Final values were averages of measurements performed in duplicate or triplicate. Levels of chemokines in the conditioned medium from cultured cells and the cell lysate were determined by reference to a standard curve produced using recombinant protein. To measure apically and basally secreted chemokines, HUVEC monolayers were grown overnight in culture conditions with TNF-α and IFN-γ as described above but on fibronectin coated transwells (Cat# 3415; Corning) with a 3 μm pore size. Monolayers were washed gently and then conditioned medium was collected from upper and lower wells after 4 h. The upper well samples were diluted with medium to equalize sample volumes and chemokine concentrations were measured by ELISA.

### Staining and imaging endothelial cells

Circular cover slips (12 mm diameter) were placed in wells of a 24-well tissue culture plate and coated with recombinant human fibronectin for 2 h at 37 °C or overnight at 4 °C. Cells were grown on these cover slips for 18–20 h in complete vascular cell growth medium at 37 °C with

5% $CO_2$. Samples were fixed with 4% paraformaldehyde (Cat# 28908;Thermo Fisher Scientific), in PBS and 2% sucrose for 30 min at RT. For permeabilization, samples were incubated with 0.1% Triton X-100 for 5 min and after extensive washing, blocked with 2% BSA/PBS for 30 min at RT. Samples were incubated at room temperature for 2 h with primary mouse antibodies against human chemokines CCL2 (5 μg, Cat# MAB679) CCL5 (5 μg, Cat# MAB278), CCL7 (5 μg, Cat# MAB282), CCL8 (5 μg, Cat# MAB281), CCL20 (5 μg, Cat# AF360), CXCL9 (5 μg, Cat# MAB392) from R&D Systems or with mouse IgG$_1$ isotype control (5 μg, Cat# MAB002) or mouse IgG$_{2B}$ isotype control (Cat# MAB004) from R&D Systems, or with anti-human GOLPH2 (2 μg, Cat# PA5-30622) or rabbit IgG isotype control from Invitrogen, or with anti-human vWF (2 μg, Cat# ab6994) from Abcam. Cells were washed three times with PBS before incubating with secondary antibody Alexa Fluor 488 (1:500, Cat# A-11017), anti-rabbit secondary antibody Alexa Fluor 568 (1:500, Cat# A-11036) and Phalloidin Alexa Fluor 647 (1:400 Cat# A30107) from Invitrogen for 1 h and after washing three times with PBS, were counter-stained with DAPI (1 μg/mL Cat# 62248; Invitrogen). Images were captured using a Leica SP8 (690) fluorescence microscope. Images were obtained with 40 × objective and analyzed using Imaris software (Oxford Instruments).

### Chemokine binding to CHO cells

Biotinylated CCL2 and CCL5 were purchased from ALMAC (U.K.). Confluent CHO cells were removed from the culture flasks using 0.05% trypsin/EDTA (Cat# R001100, Thermo Fisher Scientific) and resuspended at $1 \times 10^6$ cells/mL in HBSS (Thermo Fisher Scientific) supplemented with 2% fetal bovine serum (FACS buffer). Aliquots of 100 μl of CHO cells were incubated with or without 1 μM CCL2 or 0.1 μM CCL5 for 1 h at 37 °C. After washing with FACS buffer, the cells were stained by incubating with 1 μg/mL of streptavidin-PE (Cat# 405245, Biolegend) for 45 min at 4 °C. Samples were analyzed using an LSRFortessa flow cytometer (BD Biosciences), and the data were analyzed using FlowJo software (BD Biosciences).

### Binding of chemokines to HUVECs

HUVECs were grown as a non-stimulated or TNF-α-stimulated monolayer for 18 h on circular coverslips coated with fibronectin as described above. Cells were washed twice with Vascular Cell Basal Medium (without additives) and 120 nM of chemically synthesized human CCL2 or CCL5 with site-specific biotinylation (Almac) was added to the cells for 30 min at 37 °C in 5% $CO_2$, and after washing with Vascular Cell Basal Medium, cells were incubated with Alexa Fluor 594 streptavidin (1:500, 405240; BioLegend) for an additional 30 min. Cells were washed with Vascular Cell Basal Medium once followed by two washes with PBS and fixed with 4% paraformaldehyde and counter stained with DAPI. Images were captured using a Leica SP8 (690) fluorescence microscope and analyzed using Imaris software.

### Transduction of HUVECs to express CCL2-CXCL9chimera

HUVECs were plated at $0.2 \times 10^6$ cells per well of a 6-well tissue culture plate and grown overnight and in subsequent steps in culture conditions as described above. The following day, cells were washed and left in complete vascular cell growth medium containing 8 μg/mL polybrene for 2 h before adding pReceiver-Lv-201 control and CCL2-CXCL9chimera expressing lentivirus particles (GeneCopoeia) at a multiplicity of infection of five. In pReceiver-Lv-201 containing the CCL2-CXCL9 chimera sequences, the chimeric sequences were downstream of a CMV promoter and eGFP sequences were downstream of an SV40 promoter (sequences available upon request). After culturing for 48 h, cells were washed twice with complete vascular cell growth medium to remove extracellular virus, and cells were cultured in complete vascular cell growth medium for another 24 h. GFP-positive HUVECs were isolated by FACS using a

FACSAria II Cell Sorter (BD Biosciences) and these cells were used in flow chamber assays as described above.

### CCR2 internalization assay

Peripheral blood mononuclear cells were isolated from whole blood and treated with recombinant human CCL2 (279-MC-010; R&D Systems) at 100 ng/mL, 1 μg/mL and 10 μg/mL CCL2 for 30 min at 37 °C and cells were stained with anti-human CD4-APC, CD45RO-BV605, CCR2-PE and CXCR3-PE-Cy7 as described above, except that CCL2 was maintained at the indicated concentrations during the staining. Samples were analyzed using an LSRFortessa flow cytometer (BD Biosciences), and the data were analyzed using FlowJo software (BD Biosciences).

### Statistical analysis

All data points are from distinct samples. Statistical analyses were performed using GraphPad Prism or R. Pooled experiments are represented as median or mean ± SEM, as indicated. For each experiment, statistical tests are indicated in the figure legends. For data where a Student's *t* test was used, the data passed the Shapiro-Wilk test for normality. All Student's *t* tests were two-sided. *P* values are provided in the figures.

### Reporting summary

Further information on research design is available in the Nature Portfolio Reporting Summary linked to this article.

## Data availability

All data are included in the Supplementary Information or available from the authors, as are unique reagents used in this Article, except that release of donor/participant data must be consistent with the donors'/participants' prior consent. The raw numbers for charts and graphs are available in the Source Data files whenever possible. The bulk RNA-seq and single-cell RNA-seq data from blood cells and CITE-seq data from CSF cells generated in this study have been deposited in the Gene Expression Omnibus (GEO) database under accession code GSE284426 for the blood cell data [https://www.ncbi.nlm.nih.gov/geo/query.acc.cgi?acc=GSE284426] and accession code GSE286068 for the CSF cell data. Source data are provided with this paper.

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

## Acknowledgements

We are grateful to Tracy M. Handel, UC San Diego, for helpful advice, Jean K. Lim, Icahn School of Medicine at Mount Sinai, and Philp M. Murphy, NIAID, NIH, for critical reading of the manuscript and the members of the Research Technologies Branch, NIAID, NIH, for their help with cell sorting. This research has been funded by the Division of Intramural Research of the NIAID/NIH.

## Author contributions

F.P., S.P.S., N.K., H.H.Z., S.A., F.A.O-C., and D.J.R. performed the experimental studies and data analysis. A.S., P.J.G., H.A.L., and T.G.M. performed computational analysis. S.G. and J.K. performed image analysis. T.G.M., S.P., B.B., and J.M.F. supervised the work. F.P. and J.M.F. wrote the manuscript.

## Funding

## Competing interests

The authors declare no competing interests.
