## [Peer Review file · Nature Communications]

Migration arrest and transendothelial trafficking of human pathogenic-like Th17 cells are mediated by differentially positioned chemokines

Corresponding Author: Dr Joshua Farber

Version 0:

Reviewer comments:

Reviewer #1

(Remarks to the Author)

The authors provide a specific case study where CCL2 is not presented by HUVEC but mediates TEM, while other tested chemokines are presented on the HUVEC surface and only support arrest. However, they authors previously showed MAIT use CCR2 selectively for TEM and CCR6 for arrest in the same model. Although aspects of chemokine presentation are evaluated here, extension of the findings to Th17 cells appears limited in novelty. Differential immobilization and involvement of chemokines in arrest vs transmigration have been studied for monocytes..

The detailed profiling of CCR2+ CCR6+ subset in terms of cytokines, genes etc seems largely independent of or lacking clear relevance to the cell biology study.

The study is also limited to ex vivo analyses with HUVEC. The mechanism for differential binding to or presentation by HUVEC is not identified, and whether the model results are replicated in vivo, whether they are specific to HUVEC or instead whether different venule types can present different chemokines, is not explored. Nor whether different chemokines bind to different vascular beds. For example, how do homeostatic chemokines function in the model? Or CXCL12? Many homeostatic chemokines are not expressed by EC normally, instead by local epithelia or stroma, yet still mediate lymphocyte recruitment. Particularly homeostatic chemokines like CCL10, CCL25 etc. How do these act in the current context?

The trajectory analysis could be deleted. In the presented dataset it provides a distinct clustering approach but the concept that it relates to differentiation per se (pseudo time) is inappropriate in this context since there is no reason to think the PBMC cells in the gated CCR2+ CCR6+ PBMC subset have developed through a common precursor, or indeed that they represent a differentiation sequence.

The current understanding of leukocyte-endothelial cascades and mechanisms and trans endothelial migration has evolved and recent reviews should be cited.

Reviewer #2

(Remarks to the Author)

This work is to characterize human circulating CD4 memory T cells defined by CCR2 and CCR6 expression. They performed bulk and scRNA-seq. HUVEC endothelial transmigration and cell arrest assays with CCL2 and a few other chemokines were performed. A merit of this work is characterization of the CCR6-expressing memory T cells in terms of cytokine and gene expression. Limitations include descriptive information without definitive evidence to support the conclusion about the predicted in vivo functions of the T cell subsets.

The title "Chemokine positioning determines mutually exclusive roles for their receptors in extravasation of pathogenic human T cells" is not fully supported by the data.

The intro part should be written more clearly in a succinct manner. The rationale for the study is buried in the review-like wordy text.

A major problem is that most of the results are over-stated.

“CCR2 identifies pathogenic human type 17 cells”

Pathogenic Th17 cells can be identified by functional studies not by just an RNA-seq study. Need to tone down and limit the conclusion.

“CCR6+CCR2+CD4+ T cells are the cells migrating most efficiently across inflamed endothelium”

Inflamed endothelium is literally inflamed endothelial cells in vivo. The in vitro cytokine-stimulated may partially mimic but is not really “inflamed endothelium.”

“Chemokine localization restricts and preserves CCR2 for a role in TEM”

This is not fully supported by the presented data.

“Triggering arrest and TEM are mutually exclusive activities”

This is an exaggerated conclusion. Need to tone down.

Tissue sources of T cells should be described clearly in all figure legends.

Figure 5. “CCR2 ligands are secreted but do not bind to HUVECs”

The data for CCL2 binding to HUVEC does not sound important. How about binding to real endothelial cells in human tissues?

Figure 6. Surface-bound CCL2 can induce CCR2-mediated arrest.

The significance of this work based on the artificial CCL2-CXCL9 chimeric protein is not clear when CCL2 does not bind endothelial cells.

Figure 3. “CCR6+CCR2+ are best at TEM.”

Best? Please rephrase to be scientifically meaningful.

Fig.5B. The confocal staining should include cytoplasm staining to delineate the boundary of cells that present chemokines. As performed it is difficult to know if the chemokines bind cell membrane or non-cell surface areas coated with FN. Also control staining with Alexa Fluor 594 streptavidin alone should be performed.

Reviewer #3

(Remarks to the Author)

The authors investigate the function of chemokine receptors on effector memory CD4 T cells. While several studies use chemokine receptor expression as markers for T cell subpopulation this extraordinary well written manuscript not only shows differences in the function of chemokine receptors expressed on the same cell, but also characterizes the associated gene expression profile. They report important findings showing that CCR6+/CCR2+ not activated CD4 memory T cells possess a distinct pathogenic (inflammatory) gene signature and are able to execute transendothelial migration in vitro via CCR2 while CCR6 mediates cell adhesion. Moreover, they show that these two responses mediated by the same class of chemokine receptors are mutually exclusive.

Some points could be addressed to improve the manuscript:

Page 8: The statements "which reached statistical significance in the CCR6^{high}CCR2⁻ and CCR6+CCR2+ cells (Fig. 3 D)" and "CCL20, blocking CCR5 had no effect on TEM of the CCR6+CCR2+ cells (Fig. 3 D)" appear contradictory. The statistically significant effect of Maraviroc on TEM event on the field of vision need explanation (left panel).

Figure 4: The sentence "The addition of IFN-g led to increases in arrest by the memory cells with the largest and smallest increases in the CCR6-CCR2- and CCR6+CCR2+ cells, respectively." Is confusing with regards to the figure 4A (middle panel).

Page 8: continuing, please mention in the main text which CXCR3 inhibitor was used.

Page 9: "results were the same whether..." should read "results were similar..."

Figure 5B: The punctate appearance of endogenous cell surface bound chemokine needs explanation. The optical resolution of an SP8 does not resolve membrane rafts. Please explain/comment on the observation. Similarly, why do intracellular chemokines appear in dots in some cases while are patchier for CCL2 or CCL8. Some counterstaining for intracellular organelles would be useful. Are chemokines associated with von Willebrand factor positive Weibel-Palade bodies?

Figure 6: The surface staining of the chimeric chemokine appears more diffused (compared with figure 5B)?

Figure 5C/S4/discussion: Several publications have shown (recently the Handel lab PubMed 36719944 and previously the Thelen lab PubMed 22615942) that CCR2 acts as scavenger during cell migration, delivering the ligand for degradation, and recycling to the plasma membrane. In vitro maximum migration (bell shape curve) is achieved at low concentrations (~10 ng/ml) close to the kd for CCL2. This property of CCR2 might contribute to the TEM stimulation of T cells creating local gradients. The possible desensitization (not internalization) of CCR2 by 100 ng/ml (Fig 5C) could be due to the uniform binding and activation of CCL2 to all surface expressed receptors. This fact could be included in the discussion.

Minor:

Figure 1, panels E and F are mislabeled (main text).

Introduction:

The only typical (canonical) receptor for CCL20 is CCR6. However, the chemokine is scavenged by the atypical receptor ACKR4.

It would be useful to mention that that data obtained with MAIT cells, were from CD8 positive cells.

Version 1:

Reviewer comments:

Reviewer #1

(Remarks to the Author)

The authors have responded adequately to most concerns and discussed those that could not be addressed.

Reviewer #2

(Remarks to the Author)

The authors have clarified the issues raised by the reviewers. This has improved the manuscript.

Reviewer #3

(Remarks to the Author)

The authors have thoroughly revised their manuscript clarifying their main message. The revised version contains substantial improvements. The authors have answered all my concerns, modified the text, included references and added new data for clarification. I have no further concerns.

November 25, 2024

Point-by-point response, Parween et al., NCOMMS-23-04994 entitled “Chemokine positioning determines specialized roles during transendothelial trafficking of human Th cells with pathogenic features”

Reviewer #1 (expert in lymphocyte trafficking):

We appreciate the reviewer’s careful reading of the manuscript, critical comments, and questions.

1. Reviewer: “The authors provide a specific case study where CCL2 is not presented by HUVEC but mediates TEM, while other tested chemokines are presented on the HUVEC surface and only support arrest. However, they authors previously showed MAIT use CCR2 selectively for TEM and CCR6 for arrest in the same model. Although aspects of chemokine presentation are evaluated here, extension of the findings to Th17 cells appears limited in novelty. Differential immobilization and involvement of chemokines in arrest vs transmigration have been studies for monocytes..

The detailed profiling of CCR2+ CCR6+ subset in terms of cytokines, genes etc seems largely independent of or lacking clear relevance to the cell biology study.”

Response: Although, as the reviewer notes, we previously reported that CCR6 and CCR2 play dedicated roles in arrest and TEM, respectively, on human MAIT cells, we believe that the current manuscript includes additional important and novel findings. Regarding human T cell trafficking and considering only the specific functions of chemokine receptors, we also report here on the activities of CCR5 and CXCR3 as arrest receptors. Most importantly, we investigate the mechanism for the receptors’ dedicated activities and present evidence supporting chemokine localization as a critical determinant. This is the major advance described in the manuscript for our understanding of chemokine receptor function on these pro-inflammatory, multi-receptor-expressing T cells. Although it is true that the activities we describe here for CCR6 and CCR2 on the CD4⁺CCR6⁺CCR2⁺ T cells is the same as we described previously for MAIT cells, and are therefore not altogether novel, together these results on very different subsets of human T cells supports the generalizability of our findings.

The question of novelty also relates to the reviewer’s point that “detailed profiling of CCR2+ CCR6+ subset in terms of cytokines, genes etc. seems largely independent of or lacking clear relevance to the cell biology study.” Although we understand that these two aspects of our work could perhaps be published separately, we felt that these cells’ biological activities as revealed in our detailed investigation of their phenotype are an integral part of their broader effector capabilities that include their ability to migrate efficiently across activated endothelial cells. These data connecting chemokine receptor expression and activities with other aspects of their effector functions is a central and novel component of the manuscript. Additionally, because the other reviewers commented favorable on including the data on the non-migratory aspects of the cells’ phenotype, we have elected to leave keep both parts of our study in the revised manuscript.

2. Reviewer: “The studies is also limited to ex vivo analyses with HUVEC. The mechanism for differential binding to or presentation by HUVEC is not identified, and whether the model results

are replicated in vivo, whether they are specific to HUVEC or instead whether different venule types can present different chemokines, is not explored. Nor whether different chemokines bind to different vascular beds. For example, how do homeostatic chemokines function in the model? Or CXCL12? Many homeostatic chemokines are not expressed by EC normally, instead by local epithelia or stroma, yet still mediate lymphocyte recruitment. Particularly homeostatic chemokines like CCL10, CCL25 etc. How do these act in the current context?"

Response: Re "differential binding to or presentation by HUVEC": The binding of chemokines to glycosaminoglycans (GAGs) is well established, and no other mechanism has been described that would account for chemokine binding to HUVEC. Nonetheless, in response to the reviewer's comment, we have used Chinese hamster ovary (CHO) cells sufficient or deficient in heparan sulfate (HS, the principal chemokine-binding GAG on cell surfaces) to investigate the differential binding of CCL5 vs. CCL2 to surface HS to test the hypothesis that differential binding of chemokines to GAGs on cell surfaces could explain our findings on HUVECs. In new data (Fig. 6c), we show that recombinant CCL5, but not CCL2, binds to HS on the surfaces of CHO cells.

Re "whether the model results are replicated in vivo, whether they are specific to HUVEC or instead whether different venule types can present different chemokines, is not explored": We have expanded to studies beyond HUVECs to human dermal microvascular endothelial cells (HDMECs) and have shown, in new Supplemental Fig. 4 that 1) the binding pattern of chemokines to HDMECs matches what we found for HUVECs, and 2) the specific roles we identified for CCR2 and the other receptors using flow chamber assays with HDMECs also matched our findings using HUVECs. These data demonstrate that our findings are not limited to HUVECs, but also hold for another, very different source of human endothelial cells. In addition, we have tried many approaches to identify chemokine localization in vivo using mice with skin inflamed by injecting IL-1 α and TNF α . Although we were able to verify induction of chemokine mRNA, we have been unable to detect chemokines convincingly in the tissue using immunofluorescence and confocal microscopy or by using immunodetection with electron microscopy, despite screening multiple anti-chemokine antibodies. Using immunohistochemistry, we were also unsuccessful in detecting chemokines in human inflamed lungs (and other tissues) from autopsies of individuals dying of COVID-19. We attribute these negative results to technical limitations in sensitivity of detection due to the affinity/avidity of the available anti-chemokine antibodies. For the human samples, delays in processing autopsy tissue may also have been contributory.

Re "Nor whether different chemokines bind to different vascular beds. For example, how do homeostatic chemokines function in the model? Or CXCL12? Many homeostatic chemokines are not expressed by EC normally, instead by local epithelia or stroma, yet still mediate lymphocyte recruitment. Particularly homeostatic chemokines like CCL10, CCL25 etc. How do these act in the current context?" In agreement with the reviewer and as written in the Discussion, it is likely that different chemokines will be produced and localize differently depending on the vascular beds and the context, potentially, in some cases, altering the roles of individual receptors for arrest versus TEM. Some homeostatic chemokines are made by endothelial cells, such as CCL21 made by the HEVs – an example of a tissue-specific venules with a unique chemokine profile. Homeostatic (or other) chemokines that are not made by endothelial cells can act in two ways: either by being produced by perivascular cells or other extravascular cells (such as CCL25 made by small intestinal epithelial cells), thereby accumulating on the abluminal side and creating a transendothelial gradient for chemotaxis, or by being transported from the abluminal to the

luminal side, for example by ACKR1, thereby functioning in arrest. Trans-cellular transport by ACKR1 is generally a mechanism for presentation of pro-inflammatory chemokines, although ACKR1 can bind CXCL12 dimers, and there is likely also ACKR1-independent transport of chemokines to endothelial cell surfaces, as has been described for CCL19 (E. S. Baekkevold et al., DOI: 10.1084/jem.193.9.1105), which is not known to bind to ACKR1. The revised Discussion includes a detailed description of how the homeostatic receptor CCR7 and one of its ligands, CCL21, may function to mediate extravasation across the HEV.

3. Reviewer: “The trajectory analysis could be deleted. In the presented dataset it provides a distinct clustering approach but the concept that it relates to differentiation per se (pseudo time) is inappropriate in this context since there is no reason to think the PBMC cells in the gated CCR2+ CCR6+ PBMC subset have developed through a common precursor, or indeed that they represent a differentiation sequence.”

Response: We take the reviewer’s point, and we have removed the trajectory analysis from the revised manuscript.

4. Reviewer: “The current understanding of leukocyte-endothelial cascades and mechanisms and trans endothelial migration has evolved and recent reviews should be cited.”

Response: We have now referenced some more recent reviews in the Introduction, including by P. Kameritsch and J. Renkawitz from 2020 (DOI: 10.1016/j.tcb.2020.06.007).

Reviewer #2 (expert in TH17 cells and cell trafficking):

We appreciate the reviewer’s careful reading of the manuscript, critical comments, and questions.

1. Reviewer: “The title “Chemokine positioning determines mutually exclusive roles for their receptors in extravasation of pathogenic human T cells” is not fully supported by the data.”

Response: We have changed the title to account for what we take to be the reviewer’s objections based on his/her other comments. New title: “Chemokine positioning determines specialized roles during transendothelial trafficking of human Th cells with pathogenic features”

2. Reviewer: “The intro part should be written more clearly in a succinct manner. The rationale for the study is buried in the review-like wordy text.”

Response: The Introduction has been shortened and, we hope, made clearer.

3. Reviewer: “A major problem is that most of the results are over-stated. “CCR2 identifies pathogenic human type 17 cells” Pathogenic Th17 cells can be identified by functional studies not by just an RNA-seq study. Need to tone down and limit the conclusion.”

Response: We have changed the wording of this heading to “CCR2 identifies human

type 17 cells with a pathogenic signature”. In the text, we have also changed “pathogenic potential” and “pathogenic capabilities” to “pathogenic features”, where the original wording could be understood to imply a claim as to function that has not been established. The human CD4⁺CCR6⁺CCR2⁺ T cells have a signature of gene/protein expression characteristic of pathogenic Th17 cells, which is based on functional studies in mice, particularly in studies of experimental autoimmune encephalomyelitis. In our new data using CITE-seq to analyze CD4⁺ T cells from the cerebrospinal fluid of patients with multiple sclerosis (MS) and control subjects, we found that CD4⁺CCR6⁺CCR2⁺ T cells cluster disproportionately with cells with such a pathogenic profile of gene expression, as well as a profile found in cells thought to be pathogenic in patients with MS. By comparison with the CCR6⁺CCR2⁺ cells from control subjects, this subgroup of cells from patients with MS showed more expression of both pathogenicity-associated genes and markers of T cell activation. These data support, but of course do not prove, that the CD4⁺CCR6⁺CCR2⁺ T cells have a pathogenic role in humans.

4. Reviewer: “CCR6+CCR2+CD4+ T cells are the cells migrating most efficiently across inflamed endothelium”

Inflamed endothelium is literally inflamed endothelial cells in vivo. The in vitro cytokine-stimulated may partially mimic but is not really “inflamed endothelium.”

Response: We have changed “inflamed endothelium” to “monolayers of cytokine-activated endothelial cells”.

5. Reviewer: “Chemokine localization restricts and preserves CCR2 for a role in TEM” This is not fully supported by the presented data.”

Response: We have changed this heading to read: “CCR2 has a restricted role in TEM consistent with ligand localization”.

6. Reviewer: “Triggering arrest and TEM are mutually exclusive activities” This is an exaggerated conclusion. Need to tone down.”

Response: We have changed this heading to read: “Triggering arrest and TEM are activities of CCR2 ligands that are, respectively, bound and not bound to HUVECs”. Elsewhere in the manuscript, including in the abstract, we have changed wording to avoid a blanket claim that these activities are mutually exclusive. In agreement with the reviewer, we surely cannot generalize to all contexts, receptors, and ligands, as is made explicit in the Discussion.

7. Reviewer: “Tissue sources of T cells should be described clearly in all figure legends.”

Response: We have now detailed the tissue source of T cells in the legends of all figures. Except for the new data analyzing cells from cerebrospinal fluid, all cells were purified from the blood of healthy donors. Additionally, the first figure of the manuscript includes the sorting strategy for the T cells from blood used throughout the experiment. This information will be included in each relevant figure legend as well.

8. Reviewer: Figure 5. “CCR2 ligands are secreted but do not bind to HUVECs” The data for CCL2 binding to HUVEC does not sound important. How about binding to real endothelial cells in human tissues?”

Response: We respectfully suggest that primary HUVECs are “real endothelial cells” that have been used extensively to study endothelial cell biology. We have also added new data using primary human dermal microvascular endothelial cells (HDMECs), which demonstrated findings like the HUVECs regarding chemokine localization and the trafficking of the T cell subsets. Nonetheless, as noted in our response to Reviewer #1, we made further attempts to address the reviewer’s request by analyzing lung tissue from individuals dying from COVID-19 using immunohistochemistry. Unfortunately, we were unsuccessful in detecting chemokines in the inflamed lungs (and other tissues) from these autopsies. We attribute these negative results to technical limitations in sensitivity of detection due to the affinity/avidity of the available anti-chemokine antibodies, and possibly also due to delays that occur in processing autopsy tissue.

9. Reviewer: “Figure 6. Surface-bound CCL2 can induce CCR2-mediated arrest. The significance of this work based on the artificial CCL2-CXCL9 chimeric protein is not clear when CCL2 does not bind endothelial cells.”

Response: The point of this experiment, as for the experiment where we pre-treated T cells with soluble CCL2 immediately before adding them to the flow chamber, was not to duplicate a physiologically relevant phenomenon, but to test the hypothesis that the cause of CCR2’s failure to contribute to T cell arrest was a deficiency of CCR2 ligand at the surface of endothelial cells under conditions of flow – in contrast to the condition for the surface-bound ligands for CCR5, CCR6 and CXCR3. Although our immunofluorescent staining of endothelial cells did not reveal surface-bound CCR2 ligands, CCR2 may still have been activated by surface ligand below the limit of the limit of detection of this assay (or even by low levels of secreted ligands in the flowing medium) and CCR2’s failure to mediate arrest may have been due to an intrinsic property of CCR2 signaling and not to an insufficiency of surface ligand. In the chimeric chemokine, CCL2 was endowed with the surface-binding properties of the arrest-mediating chemokine, CXCL9. The finding that when using endothelial cells expressing the chimera, CCR2 was now able to mediate arrest is consistent with a deficiency of surface-bound CCR2 ligand as the cause of CCR2’s previous inability to do so and demonstrates that this inability was not due to any intrinsic property of the receptor. This experiment also showed that surface CCL2 could block TEM, suggesting that a transendothelial chemokine gradient might be necessary for CCR2-mediated TEM. Although this experiment used an artificial chemokine, we believe it provided important information as to bona fide mechanisms of chemokine receptor function. We have added text to the revised manuscript to explain better the information added by this experiment (and the related experiment using cells pre-treated with a high concentration of soluble CCL2).

10. Reviewer: “Figure 3. “CCR6+CCR2+ are best at TEM.” Best? Please rephrase to be scientifically meaningful.”

Response: The title of the figure, now Figure 4, has been changed to: “Unlike other CD4⁺ T cell subgroups, CCR6⁺CCR2⁺ cells are able to undergo TEM in flow chamber assays.”

11. Reviewer: “Fig.5B. The confocal staining should include cytoplasm staining to delineate the boundary of cells that present chemokines. As performed it is difficult to know if the chemokines bind cell membrane or non-cell surface areas coated with FN. Also control staining with Alexa Fluor 594 streptavidin alone should be performed.”

Response: We have incorporated phalloidin staining in our experiments to define clearly the cell boundaries and updated Figure 5B with the revised images. The pattern demonstrates that the chemokines are cell-associated. We have also included control “staining” with Alexa Fluor 594 streptavidin alone as requested (see Fig. S4D).

Reviewer #3 (expert in chemokines/chemokine receptors):

We appreciate the reviewer’s careful reading of the manuscript, critical comments, and questions. We also very much appreciate the reviewer’s positive comments.

1. Reviewer: “Page 8: The statements "which reached statistical significance in the CCR6^{high}CCR2⁻ and CCR6+CCR2⁺ cells (Fig. 3 D)" and "CCL20, blocking CCR5 had no effect on TEM of the CCR6+CCR2⁺ cells (Fig. 3 D)" appear contradictory. The statistically significant effect of Maraviroc on TEM event on the field of vision need explanation (left panel).”

Response: We apologize for the confusion. For each treatment in (new) Fig. 4 we display on the left the numbers of cells undergoing each of the steps in the flow chamber assays and on the right the ratios of numbers of arrested/rolling cells and transmigrating/arrested cells. We show the latter, normalized data to demonstrate effects specifically on arrest or TEM, since the absolute numbers of cells arresting or transmigrating will be altered by changes in numbers of cells rolling or arresting, respectively. Fig. 4D shows that the effect of maraviroc on TEM (left panel) is due to a decrease in the number of cells arresting, and not because of an effect on TEM per se (right panels). We have revised the text to clarify both the rationale for how the flow chamber data are displayed and the effects of maraviroc in Fig. 4D.

2. Reviewer: “Figure 4: The sentence "The addition of IFN-g led to increases in arrest by the memory cells with the largest and smallest increases in the CCR6-CCR2⁻ and CCR6+CCR2⁺ cells, respectively." Is confusing with regards to the figure 4A (middle panel).”

Response: We apologize for the confusion. The differential effects on arrest among the memory cell subsets of adding IFN γ as an endothelial cell stimulus is not an important point. We have altered the text to focus on the more important observations: “The addition of IFN- γ led to increases in arrest of all the memory cell subgroups, with no effect on rolling or on the percentages of arrested cells undergoing TEM (Fig. 5 A).”

3.Reviewer: “Page 8: continuing, please mention in the main text which CXCR3 inhibitor was used.”

Response: We have done that. We used the CXCR3 inhibitor AMG487.

4. Reviewer: "Page 9: "results were the same whether..." should read "results were similar..."".

Response: We have made this change.

5. Reviewer: "Figure 5B: The punctate appearance of endogenous cell surface bound chemokine needs explanation. The optical resolution of an SP8 does not resolve membrane rafts. Please explain/comment on the observation. Similarly, why do intracellular chemokines appear in dots in some cases while are patchier for CCL2 or CCL8. Some counterstaining for intracellular organelles would be useful. Are chemokines associated with von Willebrand factor positive Weibel-Palade bodies?"

Figure 6: The surface staining of the chimeric chemokine appears more diffused (compared with figure 5B)?"

Response: The reviewer raises interesting questions regarding the processing and positioning of chemokines in endothelial cells. As noted in the revised manuscript, inhomogeneity in the distribution of surface-bound chemokine has been previously described. However, studies investigating patterns of chemokine surface distribution and the bases for these patterns are limited. We have noted these patterns and discussed the relevant published data in the revised manuscript. As suggested by the reviewer, we have co-stained HUVECs and HDMECs for chemokines and GOLPH2 to identify the Golgi apparatus and co-stained HUVECs for chemokines and von Willebrand factor to identify Weibel-Palade bodies (Supplemental Figs. 6 and 7). Where there was peri-nuclear chemokine staining, it was localized to the Golgi apparatus. The chemokines we stained did not co-localize with von Willebrand factor. Regarding the surface staining for the chimeric chemokine in new Figure 7, to our eyes it appears punctate and not substantially different from the staining pattern for CXCL9 on HUVECs or HDMECs.

6. Reviewer: "Figure 5C/S4/discussion: Several publications have shown (recently the Handel lab PubMed 36719944 and previously the Thelen lab PubMed 22615942) that CCR2 acts as scavenger during cell migration, delivering the ligand for degradation, and recycling to the plasma membrane. In vitro maximum migration (bell shape curve) is achieved at low concentrations (~10 ng/ml) close to the kd for CCL2. This property of CCR2 might contribute to the TEM stimulation of T cells creating local gradients. The possible desensitization (not internalization) of CCR2 by 100 ng/ml (Fig 5C) could be due to the uniform binding and activation of CCL2 to all surface expressed receptors. This fact could be included in the discussion."

Response: We thank the reviewer for these insights. We agree that the scavenging of ligand by CCR2 could contribute to the formation of a chemotactic gradient that would drive TEM. We have added this point to the Discussion with appropriate references. We also agree that desensitization of CCR2 likely led to the inhibition of TEM in the flow chamber experiments using 100 ng/ml of CCL2, and this has now been stated in the Results.

7. Reviewer: "Minor:
Figure 1, panels E and F are mislabeled (main text).

Response: Our apologies. This has been corrected.”

8. Reviewer:”Introduction:

The only typical (canonical) receptor for CCL20 is CCR6. However, the chemokine is scavenged by the atypical receptor ACKR4.”

Response: Thank you for pointing this out. This fact has been added to the introduction

9. Reviewer: “It would be useful to mention that that data obtained with MAIT cells, were from CD8 positive cells.”

Response: In the revised manuscript we have substituted “CD8 α^+ MAIT cells” for “MAIT cells” where “CD8 α^+ ” had been omitted.